# Influence of static disorder of charge transfer state on voltage loss in organic photovoltaics

Jun Yan [1✉], Elham Rezasoltani[1], Mohammed Azzouzi[1], Flurin Eisner[1] & Jenny Nelson [1✉]

Spectroscopic measurements of charge transfer (CT) states provide valuable insight into the voltage losses in organic photovoltaics (OPVs). Correct interpretation of CT-state spectra depends on knowledge of the underlying broadening mechanisms, and the relative importance of molecular vibrational broadening and variations in the CT-state energy (static disorder). Here, we present a physical model, that obeys the principle of detailed balance between photon absorption and emission, of the impact of CT-state static disorder on voltage losses in OPVs. We demonstrate that neglect of CT-state disorder in the analysis of spectra may lead to incorrect estimation of voltage losses in OPV devices. We show, using measurements of polymer:non-fullerene blends of different composition, how our model can be used to infer variations in CT-state energy distribution that result from variations in film microstructure. This work highlights the potential impact of static disorder on the characteristics of disordered organic blend devices.

---

[1] Department of Physics and Centre for Processable Electronics, Imperial College London, London, UK. ✉email: j.yan17@imperial.ac.uk; jenny.nelson@imperial.ac.uk

A charge-transfer (CT) state at a donor–acceptor (D–A) interface in an organic photovoltaic (OPV) device is an intermediate state present after a charge transfer transition in which the electron (on the acceptor) and the hole (on the donor) reside on either side of the interface[1–6]. The properties of this CT-state (such as its energy and the reorganization energies associated with its transitions) have been shown to largely determine the open-circuit voltage loss ($V_{loss}$) of OPV devices[7,8], which is defined by $V_{loss} = E_g/q - V_{oc}$, where $V_{oc}$ is the open-circuit voltage of the solar cell, $E_g$ is its optical gap and $q$ is the elementary charge. A detailed understanding of CT-state properties is therefore necessary in order to minimise avoidable energy losses.

As a result of the intrinsic disorder in molecular conformation and packing in organic semiconductors, the energies of CT states within a given OPV blend will vary giving rise to a distribution of CT-state energies that is referred to as static disorder[4,9–12]. This static disorder gives rise to a broadening of spectral features related to the CT states. At the same time, the strong electron–phonon coupling (EPC) experienced by molecular semiconductors broadens the spectral signature of any transition at finite temperature, by an amount that is controlled by the low-frequency reorganisation energy. This type of broadening is referred to as dynamic disorder[13]. Whilst dynamic disorder is temperature ($T$) dependent, static disorder is not and T-dependent CT absorption or emission measurements have therefore often been used to differentiate them[11,14–17]. Using T-dependent measurements, recent studies by Ortmann, Vandewal, Deibel and co-workers[14,16,18] have suggested that static disorder is less important in OPV than dynamic disorder, but other studies have observed the opposite[9,11,15,17]. Although this debate is still ongoing, there is a large amount of computational and experimental evidence that the CT-state energy is sensitive to the microstructure, composition, and interfacial properties of organic semiconductor blends[19–31], and multiple CT states have also been observed in several studies[26,29,32,33]. Moreover, experimental evidence of static disorder influencing the voltage loss of OPVs has been reported[34–37]. We therefore suggest that static disorder should be considered in the analysis of voltage loss mechanisms.

To date, however, models[7,8,38] attempting to explain the link between CT-state properties and voltage loss mechanisms (via radiative and non-radiative recombination pathways) have primarily focussed on incorporating only a single CT state, whose spectral signature is broadened only by the EPC mechanism[39]. In the simplest case, the rates of absorption and emission are assumed to be governed by the high-temperature limit of non-adiabatic Marcus theory. Then, an effective value for the CT-state energy, $E_{CT,eff}$ and an effective reorganization energy for the CT to ground-state transition $\lambda_{eff}$ are obtained by fitting the reduced-absorption (or external quantum efficiency (EQE)) and -emission spectrum to the forms $\exp\left[-\frac{(\hbar\omega - E_{CT,eff} \pm \lambda_{eff})^2}{4\lambda_{eff}k_BT}\right]$, where is the, $k_B$ is Boltzmann's constant, $\hbar\omega$ is the photon energy and $\lambda_{eff}$ takes the minus sign for absorption and the plus sign for emission. In this approximation, each of the reduced spectra has a gaussian shape with a breadth of $\sqrt{2\lambda_{eff}k_BT}$, and the line-shape broadening is thus controlled by the reorganization energy associated with the CT to ground-state transition[39]. We will refer to this approach as the single-state model. The properties of this CT state are assumed to be representative of all CT states in the system, and static disorder is not explicitly considered. This approach, using either the high-temperature limit summarised above or an intermediate-temperature model based on Marcus–Levich–Jortner (MLJ) theory[7,8] has been widely used to relate CT-state energy to trends

in voltage loss in OPV systems[1,7,39–41]. It offers a way to relate $V_{oc}$ losses to other CT-state properties such as oscillator strength[8], vibrational mode energy[7,8], static dipole moment[8], and hybridization with local exciton (LE) states[38]. However, as we shall show, those relationships would change if the CT-state energy varies within the studied system. A more realistic model should also account for the inherent static disorder present in OPV, which leads to a distribution in the energy of CT states rather than a unique CT-state energy. A model that includes static disorder would allow us to account for the impact on voltage losses of phenomena that are not currently included, such as micro-structure and conformational disorder of the molecules. Several studies to date have incorporated static disorder in order to model the luminescence behaviour[11,17,42] of OPV. Those models are however all limited to a gaussian distribution of CT states (CT-DoS) and therefore cannot be applied to the general cases where a number of different CT states are present;[26,29,32,33] moreover, these models usually (ref. [11,17,42]) fail to obey the principle of detailed balance (see Supplementary Fig. 1 and Supplementary Note 1).

In this contribution, we introduce a model that incorporates static disorder in CT-state energy by including a general distribution of electronic CT states $g(E_{CT})$. Using our model, we first show that the principle of detailed balance is obeyed regardless of the shape of $g(E_{CT})$, and that use of single CT-state analysis to quantify emission and absorption can lead to incorrect results for voltage losses when static disorder is not considered. We test our model using two series of poly(3-hexylthiophene-2,5-diyl) (P3HT): non-fullerene acceptors (NFAs) blends as a function of NFA content, in which we observe the presence of two distinct CT-state features that we assign to the presence of semi-crystalline and amorphous phases of P3HT. We demonstrate that using our model, we can successfully reproduce the observed experimental changes in the absorption and emission spectra as well as in the voltage behaviour of the devices.

## Results

**Static disorder model.** Figure 1a, b illustrates a general energetic distribution of CT states, $(E_{CT})$, that might result at a D–A heterointerface in which a donor domain is surrounded by acceptor domains of different sizes and strength of interaction with the donor[25]. In general, such a distribution is likely to contain a number of CT-state manifolds, each centred at $E_{CT,Ct}$, where $t$ denotes the order of CT manifold. We thus describe the static disorder in the system by a normalised distribution function for CT-state energies given by

$$g(E_{CT}) = \sum_t c_t D_t(E_{CT}) \qquad (1)$$

where $c_t$ is a constant weighting coefficient such that $\sum_t c_t = 1$, and $D_t(E_{CT})$ is a line-shape function. Practically, $D_t(E_{CT})$ can be any function. In our model we use a gaussian line-shape for $D_t(E_{CT})$, i.e., $D_t(E_{CT}) = \frac{1}{\sigma_{CT,t}\sqrt{2\pi}}\exp\left[-\frac{1}{2}\left(\frac{E_{CT}-E_{CT,Ct}}{\sigma_{CT,t}}\right)^2\right]$, where $\sigma_{CT,t}$ is the width of the individual gaussian function. In the calculations shown below, each gaussian function is integrated in the range of $[E_{CT,Ct} - 5\sigma_{CT,t}, E_{CT,Ct} + 5\sigma_{CT,t}]$, ensuring $\int_a^b D_t(E_{CT})d(E_{CT}) = 1$, hence $\int g(E_{CT})d(E_{CT}) = 1$, where $a = E_{CT,Ct} - 5\sigma_{CT,t}$ and $b = E_{CT,Ct} + 5\sigma_{CT,t}$.

As in previous models[7,8,38], we assume that radiative and non-radiative recombination occur only via the CT states, either directly after exciton dissociation or by reformation of the CT state from free charges. The radiative and non-radiative CT-to-ground-state transitions occur between vibronic modes of each state and are accompanied by the emission of a photon or of

several vibrational quanta, respectively[43,44]. We additionally assume that absorption, emission and non-radiative recombination transitions occur in the weak coupling limit so that they can be described by non-adiabatic Marcus theory, and we invoke the Franck Condon principle to include transitions between different vibrational modes, as illustrated in Fig. 1c. Here, we adopt the method introduced by Jortner to describe the rate constant of transition between the CT state and the ground state[45], in which we distinguish between high and low-frequency vibronic modes. We consider that the states in each CT-state manifold (denoted as $t$ in the subscript) share the same set of parameters, such as oscillator strength ($f_{osc,t}$) and low-frequency reorganization energies ($\lambda_{o,t}$), except for their energies ($E_{CT}$). We assume quasi-thermal equilibrium (QTE) conditions, meaning that the occupation function of each electronic CT state should be considered in the expression for recombination, and that state occupation should follow Boltzmann statistics. Therefore, the total rate constant ($K_{nr}$) (s$^{-1}$) of non-radiative recombination from the CT state can be expressed as the sum of the contribution from all CT manifolds, as:

$$K_{nr} = \frac{1}{Z_{rec}} \sum_t \int_a^b \frac{2\pi}{\hbar} V_t(E_{CT})^2 \text{FCWD}_{rec,t}(0, E_{CT}) c_t D_t(E_{CT})$$
$$\exp\left(-\frac{E_{CT}}{k_B T}\right) d(E_{CT}) \qquad (2)$$

where $V_t(E_{CT})$ is the electronic coupling between CT and ground state described by the generalized Mulliken–Hush method[46,47]. FCWD$_{rec,t}$ is the Franck–Condon-Weighted Density of States (FCWD) for recombination for CT-state manifold $t$. The radiative recombination ($k_{abs}(\hbar\omega)$) (s$^{-1}$ eV$^{-1}$) and absorption rate constants per photon energy ($k_r(\hbar\omega)$) (s$^{-1}$ eV$^{-1}$) can be expressed as the sum of the contribution from all CT manifolds using a similar expression and all depend on the FCWD[8], via

$$k_{abs}(\hbar\omega) = \frac{1}{Z_{abs}} \sum_t \int_a^b \frac{W}{3\pi\epsilon_0\hbar^4} \left(\frac{\hbar\omega}{c}\right)^3 M_t(E_{CT})^2 \text{FCWD}_{abs,t}$$
$$(\hbar\omega, E_{CT}) c_t D_t(E_{CT}) d(E_{CT}) \qquad (3)$$

$$k_r(\hbar\omega) = \frac{1}{Z_{rec}} \sum_t \int_a^b \frac{1}{3\pi\epsilon_0\hbar^4} \left(\frac{\hbar\omega}{c}\right)^3 M_t(E_{CT})^2 \text{FCWD}_{rec,t}$$
$$(\hbar\omega, E_{CT}) c_t D_t(E_{CT}) \exp\left(-\frac{E_{CT}}{k_B T}\right) d(E_{CT}) \qquad (4)$$

where $W$ is the photon density and accounts for the strength of electro-magnetic field around the molecule[48]; $\epsilon_0$ is the permittivity of the free space; $M_t(E_{CT})$ is the transition dipole moment for CT manifold $t$ and is related to the oscillator strength of the CT manifold ($f_{osc,t}$) under the dipole approximation[49,50] via $M_t(E_{CT}) = \sqrt{(3/2)q^2\hbar^2 f_{osc,t}/(E_{CT}m_e)}$, with $m_e$ the electron mass and $q$ the elementary charge. $K_r$ (s$^{-1}$) is obtained by integrating Eq. (4) over $\hbar\omega$, via

$$K_r = \int_0^\infty k_r(\hbar\omega) d\hbar\omega \qquad (5)$$

For CT-state manifold ($t$), the FCWD for absorption (FCWD$_{abs,t}(\hbar\omega, E_{CT})$) and recombination (FCWD$_{rec,t}(\hbar\omega, E_{CT})$) follow MLJ theory[7,8], and can be expressed as

$$\text{FCWD}_{abs,t}(\hbar\omega, E_{CT}) = \frac{1}{\sqrt{4\pi\lambda_{o,t}k_B T}} \times \sum_{m=0}^\infty \sum_{n=0}^\infty \frac{e^{-S_t}S_t^{n-m}m!}{n!} \left[(L)_m^{n-m}(S_t)\right]^2$$
$$\exp\left[-\frac{(-\hbar\omega + E_{CT} + \lambda_{o,t} + (n-m)\hbar\Omega_t)^2}{4\lambda_{o,t}k_B T}\right] \exp\left(-\frac{m\hbar\Omega_t}{k_B T}\right) \qquad (6)$$

$$\text{FCWD}_{rec,t}(\hbar\omega, E_{CT}) = \frac{1}{\sqrt{4\pi\lambda_{o,t}k_B T}} \times \sum_{m=0}^\infty \sum_{n=0}^\infty \frac{e^{-S_t}S_t^{n-m}m!}{n!} \left[(L)_m^{n-m}(S_t)\right]^2$$
$$\exp\left[-\frac{(\hbar\omega - E_{CT} + \lambda_{o,t} + (n-m)\hbar\Omega_t)^2}{4\lambda_{o,t}k_B T}\right] \exp\left(-\frac{m\hbar\Omega_t}{k_B T}\right) \qquad (7)$$

where $\lambda_{o,t}$ and $\lambda_{i,t}$ are the low-frequency and high-frequency reorganization energy for CT manifold $t$, respectively, $S_t = \lambda_{i,t}/\hbar\Omega_t$ is the Huang Rhys factor[51], $\hbar\Omega_t$ is the averaged harmonic energy spacing, typically 0.15–0.20 eV for molecules made of many carbon–carbon bonds[7,8] and $m$ and $n$ are the quantum

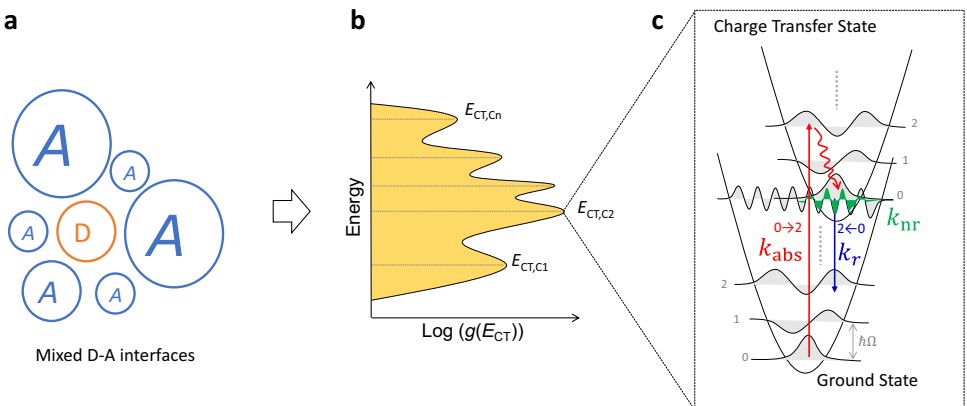

**Fig. 1 Illustration of mixed D–A interfaces, $g(E_{CT})$ distribution, absorption and recombination mechanism. a** Mixed D–A interfaces are displayed as a donor domain (orange) surrounded by different sizes and distances of acceptor domains (blue) on the left, resulting in **b** a random energetic distribution of $g(E_{CT})$ with multiple sub-CT manifolds. **c** The potential energy surface diagram shows the radiative (blue arrows) and non-radiative (green shaded area) recombination, and absorption (red straight arrow) for each CT state. Only the strongest transition for absorption (→ 2) and emission (2 ← 0) (radiative recombination) is displayed. Curved red arrow depicts the ultrafast vibrational thermalization process after photoexcitation in each electronic CT state. We neglect triplet CT states (CT$^3$) since they have been shown to have a negligible effect in some OPV systems[72,73].

numbers of the vibrational mode of the initial and final state, respectively. $L_m^{n-m}(S_t)$ is then the generalized Laguerre polynomial of degree $m$[8]. The factor $Z_{abs}$ and $Z_{rec}$ in Eqs. (2–4) are the partition functions, and are defined as the sum of occupation of vibrational CT states for absorption, and as the sum of occupation of both vibrational and electronic CT states for recombination, and are given by

$$Z_{abs} = \sum_t \sum_{m=0}^{\infty} \exp\left(-\frac{m\hbar\Omega_t}{k_B T}\right) \tag{8}$$

$$Z_{rec} = \sum_t \left\{ \sum_{m=0}^{\infty} \exp\left(-\frac{m\hbar\Omega_t}{k_B T}\right) \int_a^b c_t D_t(E_{CT}) \exp\left(-\frac{E_{CT}}{k_B T}\right) d(E_{CT}) \right\} \tag{9}$$

When the absorption and emission spectra are constructed from these FCWD distributions, the spectra represent the sum of contributions from different CT-state energies, and the resulting absorption and emission bands are broadened not only by the low-frequency broadening of each vibronic line and the FC vibronic progression, but also by the distribution of CT-state energies. Attempts to interpret such spectra in terms of a single CT-state energy would lead to incorrect estimates for both the CT-state energy and the reorganisation energies.

In QTE, the volumetric radiative ($R_r$) and non-radiative ($R_{nr}$) recombination rates of the system when under optical or electrical bias are controlled by the chemical potential energy, $\mu$, of the system. $\mu$ is defined as the free energy of CT-state excitons in the biased system and quantifies the disturbance from equilibrium (where $\mu = 0$). In QTE all CT states in the ensemble share the same chemical potential energy ($\mu$) and $R_r, R_{nr}$ are amplified from their equilibrium values by the factor $\exp(\frac{\mu}{k_B T})$ such that

$$R_{nr} = K_{nr} n_{CT0} \times \exp\left(\frac{\mu}{k_B T}\right) \tag{10}$$

$$R_r = K_r n_{CT0} \times \exp\left(\frac{\mu}{k_B T}\right) \tag{11}$$

where $n_{CT0}$ is the density of CT states in thermal equilibrium. In QTE, the recombination rate constants ($K_r, K_{nr}$) and emission spectra shape ($k_r$) do not change with $\mu$. However in cases where QTE is no longer valid, for example, if the spatially separated states are not strongly coupled, as explored by Melianas et al.[52], we would, in general, expect different chemical potential energies at different interfacial sites. Then the above expression for emission would need to be adapted as we explore in Supplementary Note 5. The full model describing the voltage losses as well as the rate constants of the radiative and non-radiative transitions with static disorder included is presented in Supplementary Methods 2–4.

In the following sections, we first investigate the impact of static disorder on emission, absorption, and voltage losses analysis from a theoretical point of view, using the model developed here. We then apply the model to investigate two experimental P3HT:NFA systems, where static disorder is evident from the CT-state emission spectra and can be related to the phase behaviour of the blends.

## General model results

The model presented in the previous section offers a way to incorporate static disorder into a model of radiative and non-radiative recombination via the CT state with a general CT-DoS shape. We now explore the consequences of incorporating static disorder into the model, firstly on the absorption and emission profile and hence on radiative and non-radiative CT-state recombination rate constants, then on voltage

and efficiency losses, and finally on the validity of single-state analysis when static disorder is present.

We firstly explore three variations of $g(E_{CT})$, as shown in Fig. 2a,e, i, a single gaussian CT manifold with varied distribution width (Fig. 2a, DoS 1), two gaussian CT manifolds with varied distribution width (Fig. 2e, DoS 2), and two gaussian CT manifolds with varied peak energy of the lower energy CT manifold (Fig. 2i, DoS 3). As a first step, we show that our model obeys the principle of detailed balance, which links the absorption and emission in a solar cell through the ambient black body radiation flux ($\phi_{BB}$)[53], and which has been shown to be obeyed in most inorganic[54–56] and organic[57,58] semiconductor based photovoltaics. The equations underpinning detailed balance are given by Supplementary Equation (17)[53]. Shown in Fig. 2b, f, j is the calculated and normalized absolute emission rate constant ($k_r$) per unit photon energy ($\hbar\omega$), the absorptance ($A$), and the ratio $k_r/\phi_{BB}$, for three types of $g(E_{CT})$). The perfect overlap between the tail of $A$ and $k_r/\phi_{BB}$ in the presence of static disorder demonstrates the validity of the detailed balance principle using our model. The parameters used to produce Fig. 2 are presented in Supplementary Tables 1–3, and we note here that neither the choice of the values of those parameters nor the shape of $g(E_{CT})$ should violate the principle of detailed balance using our model (see also Supplementary Fig. 2).

We now analyse the effect of different levels of static disorder (through varying $g(E_{CT})$) on the absorption and emission profiles of photovoltaics and finally on the voltage losses (Fig. 2). We carry out a simulation assuming that all the other CT-state parameters are preserved while changing $g(E_{CT})$ using a typical set of CT-state parameters (given in Supplementary Tables 1–3). We note that this assumption is unlikely to hold in real devices, but it is a useful and necessary first approximation to model the effect of static disorder. The effects of increasing static disorder on the absorption and emission profiles of the device are shown in Fig. 2b, f, j. The model shows that the low-energy part of $A(\hbar\omega)$ becomes broader with broadened $g(E_{CT})$, whilst the peak of the emission spectrum shifts to the red. This is reflected in the total (integrated over energy) rate constant of radiative recombination ($K_r$, Fig. 2c, g, k), which decreases slightly with broadened $g(E_{CT})$, in accordance with the energy gap law, which is embodied in our model. Concomitantly, as the CT states spread out in energy, the proportion of non-radiative transition from lower energy CT state to ground state is increased, resulting in higher values of the energy-integrated non-radiative rate constant $K_{nr}$ as also shown in Fig. 2c, g, k. From the rate constants of radiative and non-radiative recombination, we can then calculate the voltage loss due to non-radiative recombination, $\Delta V_{nr}$, which is the main voltage loss contribution to the voltage loss in OPV[39,58]. Clearly, as shown in Fig. 2d, h, l, increased static disorder (i.e., as $g(E_{CT})$ is broadened) is detrimental to both $\Delta V_{nr}$, which increases and $V_{oc,rad}$, which decreases. $V_{oc,rad}$ is the radiative limit of $V_{oc}$ and represents the maximum open-circuit potential available for a device of given absorption profile and, since it can be defined precisely for any absorption spectrum, is a more useful quantity than $E_g$[59] and $E_{CT}$[13]. The decreased $V_{oc,rad}$ with increasing $\sigma_{CT}$ can be rationalized by the broadened absorption edge, which leads to more radiative losses according to the reciprocity relation[53], while increased $\Delta V_{nr}$ with $\sigma_{CT}$ is caused by enlarged $K_{nr}$ and reduced $K_r$ (Supplementary Equation (3) and (4)). This leads to a negative dependence of $\Delta V_{nr}$ on $V_{oc,rad}$, as shown in Fig. 3a for the three types of $g(E_{CT})$ considered, where $\Delta V_{nr}$ is plotted against $V_{oc,rad}$[8,38,58]. We also note here that the extra non-radiative voltage losses caused by static disorder that were predicted by Burke et al.[9] (i.e., $\sigma_{CT}/2k_B T$) using a gaussian distribution of CT states and without considering the energy gap

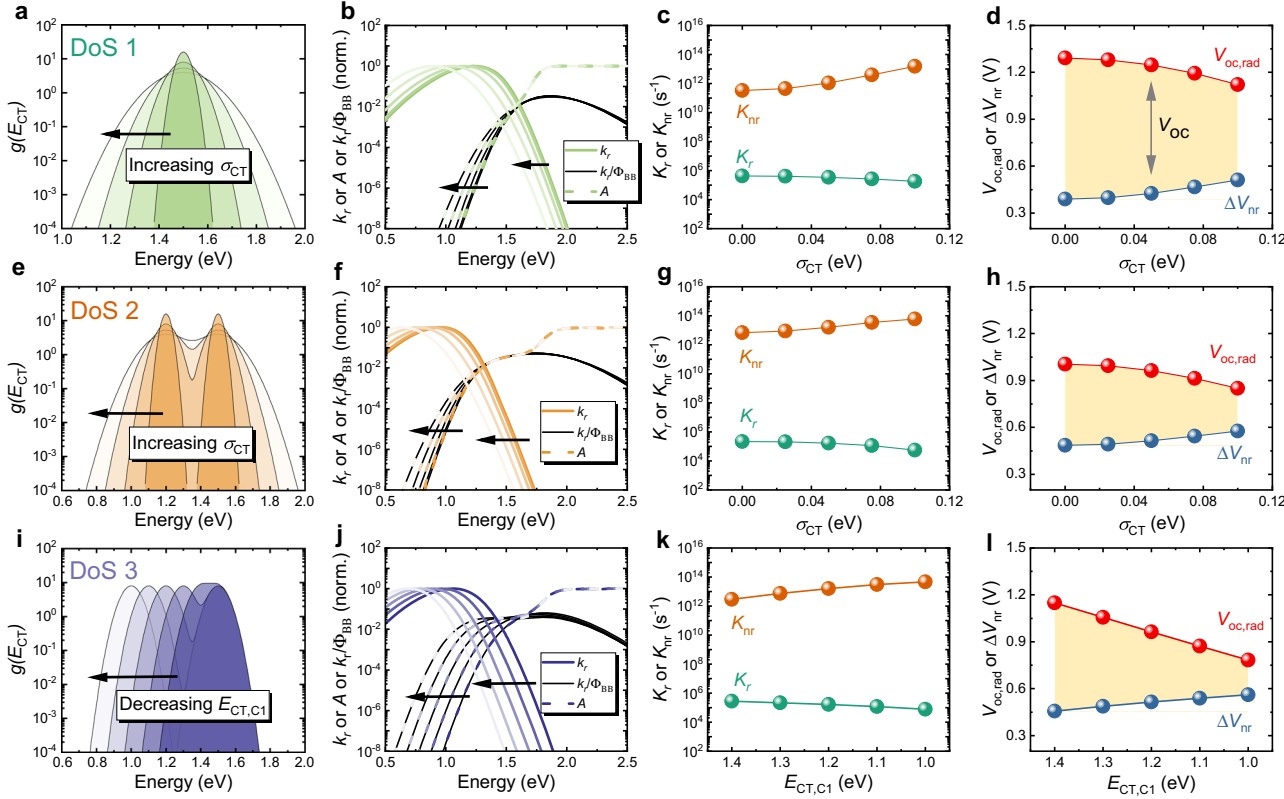

**Fig. 2 General model results on the effect of static disorder.** Three types of $g(E_{CT})$ are shown: **a** DoS 1: Single gaussian CT-state manifold with varied $\sigma_{CT}$; **e** DoS 2: Two gaussian CT manifolds with varied $\sigma_{CT}$ (two CT manifolds have the same $\sigma_{CT}$ and varies simultaneously); **i** DoS 3: Two gaussian CT manifolds with $E_{CT,C1}$ (lower energy CT manifold) changed while $E_{CT,C2}$ was fixed. Normalized rate constant per photon energy of emission ($k_r$), absorptance ($A$) and $k_r/\phi_{BB}$ for (**b**) 1 gaussian CT states with varied $\sigma_{CT}$; **f** 2 Gaussian CT manifolds with varied $\sigma_{CT}$; **j** 2 gaussian CT manifolds with varied $E_{CT,C1}$. Rate constant of radiative ($K_r$) and non-radiative ($K_{nr}$) transition for (**c**) 1 gaussian CT states with varied $\sigma_{CT}$; **g** 2 gaussian CT manifolds with varied $\sigma_{CT}$; **k** 2 gaussian CT manifolds with varied $E_{CT,C1}$. $\Delta V_{nr}$ and $V_{oc,rad}$ for (**d**) 1 gaussian CT states with varied $\sigma_{CT}$; **h** 2 gaussian CT manifolds with varied $\sigma_{CT}$; **l** 2 gaussian CT manifolds with varied $E_{CT,C1}$. The range of $\sigma_{CT}$ is chosen based on experimentally observed values ranging up to ~0.1 eV[9, 11]. Yellow shaded area indicates the size of $V_{oc}$ determined by $V_{oc,rad} - \Delta V_{nr}$. The detailed parameters used in this figure are listed in Supplementary Tables 1–3.

law are overestimated as compared to our model results (see Supplementary Fig. 3).

Using the single gaussian CT-state distribution as an example, we now go on to model the effects of different CT-state parameters that affect CT-state recombination as a function of static disorder, as in our previous work;[8] the effects of varying high- and low-frequency reorganization energies ($\lambda_i$ and $\lambda_o$), the CT-state to ground-state oscillator strength ($f_{osc}$), $E_{CT}$, and the ratio of CT to LE state densities ($N_{CT/LE}$) on $\Delta V_{nr}$ and $V_{oc,rad}$ are thus plotted in Fig. 3b, for two different $\sigma_{CT}$, i.e., $\sigma_{CT} = 0$ and $\sigma_{CT} = 0.1\,eV$ (further details are in Supplementary Figs. 4–8). The results show that the effect of each of the parameters (for further explanation refer to Ref. [8]) is qualitatively preserved regardless of the level of static disorder present in the device, since changing $\sigma_{CT}$ from 0 to 0.1 eV, as shown in Fig. 3b, shifts the whole plot to higher $\Delta V_{nr}$ and lower $V_{oc,rad}$. However, the magnitude of the effect of changing each of the other parameters changes at different $\sigma_{CT}$, i.e., increasing $\sigma_{CT}$ lowers the impact of $\lambda_o$ and $\lambda_i$, but enlarges the impact of $f_{osc}$, $E_{CT}$, and $N_{CT/LE}$ (see Supplementary Figs. 4–8 for details). The lower impact of reorganisation energy suggests that if static disorder is dominant, the impact of dynamic disorder is then reduced. Clearly, static disorder can have a major impact on the analysis of voltage loss. We also compare our model with the trend in voltage loss predicted by Benduhn et al.[7] with experimental data taken from ref. [60]. as well as data collected in this work (Fig. 3b). It is evident

that the introduction of static disorder influences the relationship between the material parameters and $\Delta V_{nr}$ and $V_{oc,rad}$. We further discuss the effects of static disorder on the voltage and efficiency limit in Supplementary Note 2, noting here that increasing $\sigma_{CT}$ from 0 to 0.1 eV can affect the relationship between $V_{oc}$ and $E_g/q$ (hence PCE vs. $E_g/q$ relation) (see Fig. 3c, d), and reduce both the voltage and efficiency by ~33% according to our model.

We now comment on the validity of single-state analysis when static disorder is present. Experimental emission and absorption spectra of OPV devices, plotted in a similar way to that shown in Fig. 2b, f, j have often been analysed using single-state model based on semi-classical Marcus theory for transitions from or to a single excited state[39]. Here, we wish to determine whether we can use single-state model analysis to reproduce the correct emission, absorption and voltage loss for a system in which static disorder is present, using the simulated emission and absorption spectra in Fig. 2b for the analysis. This offers a way of testing the validity of single-state model analysis theoretically. Using data from Fig. 2b and the relationships for a single CT state, we can extract $E_{CT,eff}$ and $\lambda_{eff}$ as shown in Fig. 3e, f. We first notice that $E_{CT,eff}$ decreases while $\lambda_{eff}$ increases upon increasing static disorder, which leads to an incorrect interpretation of the spectrum since the centre of the CT-state manifold ($E_{CT,C}$) and the associated low-frequency reorganization energy ($\lambda_o$) are constant in the

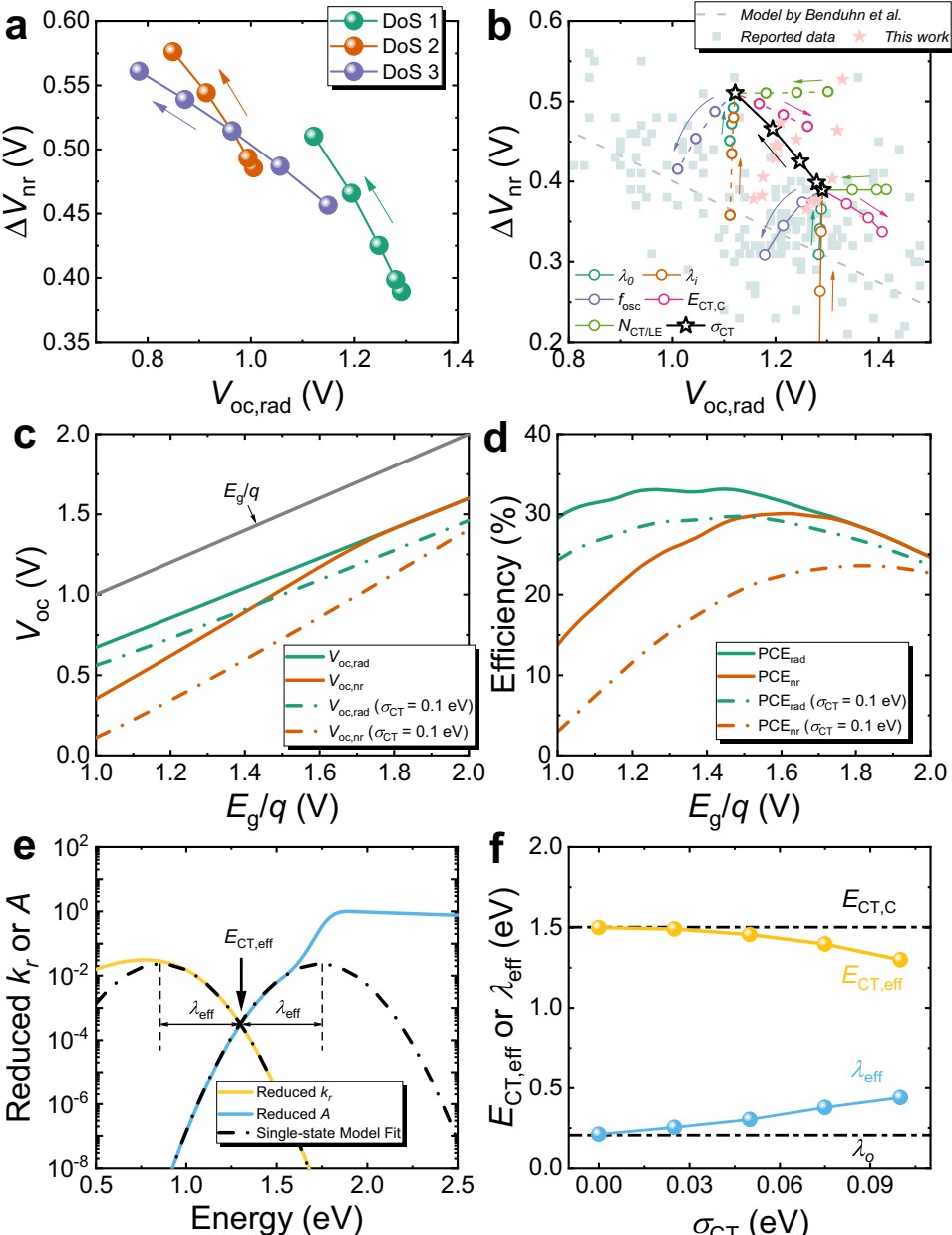

**Fig. 3 Voltage loss analysis and model comparisons, voltage and efficiency limits and single-state analysis. a** $\Delta V_{nr}(V_{oc,rad})$ plot generated from Fig. 2 with the three different DoS-CT distributions shown in Fig. 2, i.e., a single gaussian CT-state manifold (DOS 1), a Two gaussian CT-state manifold of varying breadth, and a two gaussian CT-state manifold of varying peak energies. Arrows indicate the direction of increasing static disorder. **b** Voltage loss model comparisons using $\Delta V_{nr}(V_{oc,rad})$ plot with varied properties of CT state, including $\lambda_o$, $\lambda_i$, $f_{osc}$, $E_{CT,C}$, $N_{CT/LE}$, and $\sigma_{CT}$. The ranges for these parameters are $\lambda_o = [0.05, 0.2]$ eV, $\lambda_i = [0.05, 0.2]$ eV, $f_{osc} = [10^{-2}, 1]$, $E_{CT,C} = [1.5, 1.65]$ eV, $N_{CT/LE} = [10^{-3}, 1]$, and $\sigma_{CT} = [0, 0.1]$ eV. Solid lines with circles represent the case with $\sigma_{CT} = 0.0eV$, while dashed lines with circles represent the case with $\sigma_{CT} = 0.1eV$. The calculations are compared with the model by Benduhn et al.[7] (dashed grey line) and a large amount of reported data taken from ref. [60] (light-grey scatters) as well as the data collected in this work (light-red stars). The arrows in (**e**) indicate the direction of increasing variable values. The details of the effect of each parameter on absorption, emission, rate constants, and $V_{loss}$ have been presented in Supplementary Figs. 4–8. The parameters used to produce Supplementary Figs. 4–8 are listed in Supplementary Table 4. Note here the model by Benduhn is plotted as function of $V_{oc,rad}$ using an approximated linear relation that $V_{oc,rad} = 0.833E_{CT}/q$ (see Supplementary Fig. 9). **c** Voltage limits and (**d**) Efficiency limits as a function of optical gap divided by elementary charge (i.e. $E_g/q$). Detailed parameters and discussion for voltage and efficiency limits can be found in Supplementary Note 2 and Supplementary Table 5. **e** Example of single-state analysis using reduced emission ($\propto k_r/(\hbar\omega)^3$) and absorption ($\propto A/\hbar\omega$) spectrum. **f** Extracted $E_{CT,eff}$ and $\lambda_{eff}$ as a function of $\sigma_{CT}$ using 1-gaussian DoS as an example. Simulated emission and absorption spectrum are used as the input for single-state analysis. Details of the comparisons on the calculated emission, absorption, recombination rate constant, and voltage losses are shown in Supplementary Fig. 11. The parameters used to produce the input spectrum are listed in Supplementary Table 1.

input spectrum. We then use the extracted values of $E_{CT,eff}$ and $\lambda_{eff}$ as input for the intermediate-temperature single-state model[8] (i.e., $E_{CT} = E_{CT,eff}, \lambda_o = \lambda_{eff}, \sigma_{CT} = 0$) while the other parameters are conserved (see Supplementary Table 1) and show the simulated spectrum and the radiative and non-radiative decay constants of the CT states in Supplementary Fig. 11. The rate constants calculated using the values extracted assuming an effective single CT state show an overestimation of the non-radiative decays and an underestimation of the radiative decay rate constants relative to the true rate constants. The uneven change in the two rate constants results in a higher estimation of $\Delta V_{nr}$ than the true value obtained by considering the distribution of CT-state energies (Supplementary Fig. 11d). This observation, that the single-state analysis overestimates $\Delta V_{nr}$, also holds for other CT-DoS distributions that we have considered and is more pronounced in systems with low LE–CT-state offset (see Supplementary Figs. 12 and 13 for details). We therefore conclude that the single-state analysis is only valid when there is little or no static disorder present (see Supplementary Fig. 11). These results emphasise the importance of considering the impact of static disorder of CT states when analysing the absorption and emission spectra as well as the voltage losses in a device.

**Relating phase behaviour to static disorder.** For the purpose of testing the utility of our model, we now proceed to use it to explain the experimental results of two OPV blends, namely P3HT: ((5Z,5′Z)−5,5′-(((4,4,9,9-tetraoctyl-4,9-dihydro-s-indaceno [1,2-b:5,6-b′]dithiophene-2,7-diyl) bis(benzo[c][1,2,5]thiadiazole-7,4-diyl)) bis (methanylylidene)) bis (3-ethyl-2-thioxothiazolidin-4-one)) (O-IDTBR)[61] and P3HT: (5Z,5′Z)−5,5′-((7,7′-(6,6,12, 12-tetraoctyl-6,12-dihydroindeno[1,2-b]fluorene-2,8-diyl)bis (benzo[c][1,2,5]thiadiazole-7,4-diyl))bis(methanylylidene))bis(3-ethyl-2-thioxothiazolidin-4-one) (O-IDFBR)[62]. The chemical structures of the molecules are shown in Supplementary Fig. 14. The two materials are particularly interesting because they are chemically very similar but due to the difference in the molecular planarity, the planar O-IDTBR shows an increased propensity to crystallize in blends than the twisted O-IDFBR[63]. Using detailed optical probes, we have previously shown that in blends with P3HT, with O-IDTBR the crystallinities of both P3HT and O-IDTBR are largely preserved over a wide range of blend ratios[63], whereas for the less crystalline O-IDFBR blends P3HT crystallinity is easily disrupted by introducing more O-IDFBR[63], leading to more amorphous interfaces at higher O-IDFBR content blends. The contrasting behaviour of P3HT:O-IDFBR and P3HT:O-IDTBR as a function of composition[26,32,33] allows us to fabricate two sets of devices with different underlying $g(E_{CT})$.

To understand how phase behaviour affects the $g(E_{CT})$ for the O-IDTBR and O-IDFBR devices, we first measured EQE and electroluminescence (EL) (at low injection) and present the results for devices with NFA wt% representative of different composition regimes, i.e., 20%, 40%, and 70%, in Fig. 4. Focusing first on the EQE edges, we can identify clear CT-state absorption in all the blends, as it can be clearly differentiated from the singlet absorption of the pristine donor and acceptor, as displayed in Figs. 4a, d . We also note that both the O-IDTBR and O-IDFBR devices show changed CT-state absorption (i.e., EQE edges) upon increasing NFA wt%. By estimating the slope of the EQE edges we extract the Urbach energies, as displayed in Fig. 4a, d. Interestingly, the edges of EQE spectra for blends of different wt% of O-IDTBR show roughly the same Urbach energy at 64 ± 2 meV, while the value for the O-IDFBR devices reduces from 94 ± 2 (for the 20%) to 70 ± 2 (for the 70%) meV, upon increasing O-IDFBR wt% from 20% to 70%. This suggests that the DoS-CT distribution of the lower energy CT manifold likely remains

unchanged for the O-IDTBR devices but changes for the O-IDFBR devices when NFA wt% is varied, a conclusion which is supported by the EL spectra (Fig. 4 b, e), discussed below. Further evidence of static disorder for the studied devices has also been obtained in T-dependent EQE and EL measurements, where we observed no sharpening of the EQE or EL tail with reduced temperature as shown in Supplementary Figs. 15 and 16. This is consistent with a picture in which static disorder is dominant over dynamic disorder[11].

In the O-IDTBR devices (Fig. 4b), the EL peak from CT emission stays at around 1.06 eV regardless of O-IDTBR wt%, and only the intensity changes, i.e., the intensity decreases with increasing O-IDTBR wt%. Due to the strong tendency of O-IDTBR to crystallize and the relatively low emission energy of O-IDTBR[63,64], emission from the O-IDTBR exciton is present in all O-IDTBR devices, and dominates the emission spectrum in the 70% device. In contrast, for the O-IDFBR devices (Fig. 4e), firstly the EL spectrum shows only the emission from CT states as indicated by the much lower energy EL emission from the blends relative to the PL of pristine P3HT (which has the lower optical gap of the two components in this blend), supporting the fact that O-IDFBR mixes well with P3HT[64]. Secondly, we can see two CT emission peaks (at ~1.1 and ~1.4 eV) with 20% O-IDFBR in Fig. 4e, with the relative intensity of the lower energy peak decreasing with increasing O-IDFBR wt%. Combined with the EQE analysis above, these results thus suggest that the O-IDTBR devices and O-IDFBR devices possess different underlying $g(E_{CT})$ as a function NFA content.

To visualize the distribution of $g(E_{CT})$, we further carry out EL experiments under higher injection conditions. EL peak shifting with injection current is a signature of state filling, which can only occur in the presence of static disorder[17,65]. Injection-dependent EL measurements are therefore a direct probe of the existence of static disorder. Interestingly, despite this expectation the peak EL has often been observed to be unchanged[52,65]. Nevertheless, in both O-IDTBR and O-IDFBR devices with 20% NFA wt%, we identify two sub-gap emission peaks at high injection condition, which we term $CT_1$ and $CT_2$, as labelled in Figs. 5a, d, for the lower and high energy CT state, respectively. We suggest that the two peaks are induced by interfaces with semi-crystalline polymer (lower energy $CT_1$) and interfaces with amorphous polymer (higher energy $CT_2$) in accordance with our previous findings (ref. [63]) on the phase behaviour of the two systems. We note that multiple CT manifolds have also been observed by different groups in other OPV material systems[26,32,33].

For both P3HT:O-IDTBR and P3HT:O-IDFBR with 20% wt% NFA, semi-crystalline P3HT accounts for ~70% of the polymer volume in the blend, while amorphous P3HT contributes only ~30% according to previous analysis[63]. Focusing first on O-IDTBR blends, upon increasing the injection current, in the 20% O-IDTBR blends, we see a decrease of the intensity of $CT_1$ relative to $CT_2$, while the peak position of $CT_1$ remains almost constant. This constant $CT_1$ peak position upon increasing injection is consistent with a narrow distribution of CT-state energies at the interface between semi-crystalline donor and acceptor phases. The width of the $CT_2$ manifold is notably larger than $CT_1$ indicating that it could stem from a more disordered manifold of CT states as would be found when at least one component is amorphous. We note that O-IDTBR remains semi-crystalline in the blends in all compositions[63] and therefore we suggest that $CT_1$ and $CT_2$ correspond to semi-crystalline O-IDTBR: semi-crystalline P3HT and semi-crystalline O-IDTBR: amorphous P3HT, respectively. Upon introducing more O-IDTBR in the blends, the reduced intensity of $CT_1$ can be assigned to the aggregation and crystallization of O-IDTBR and

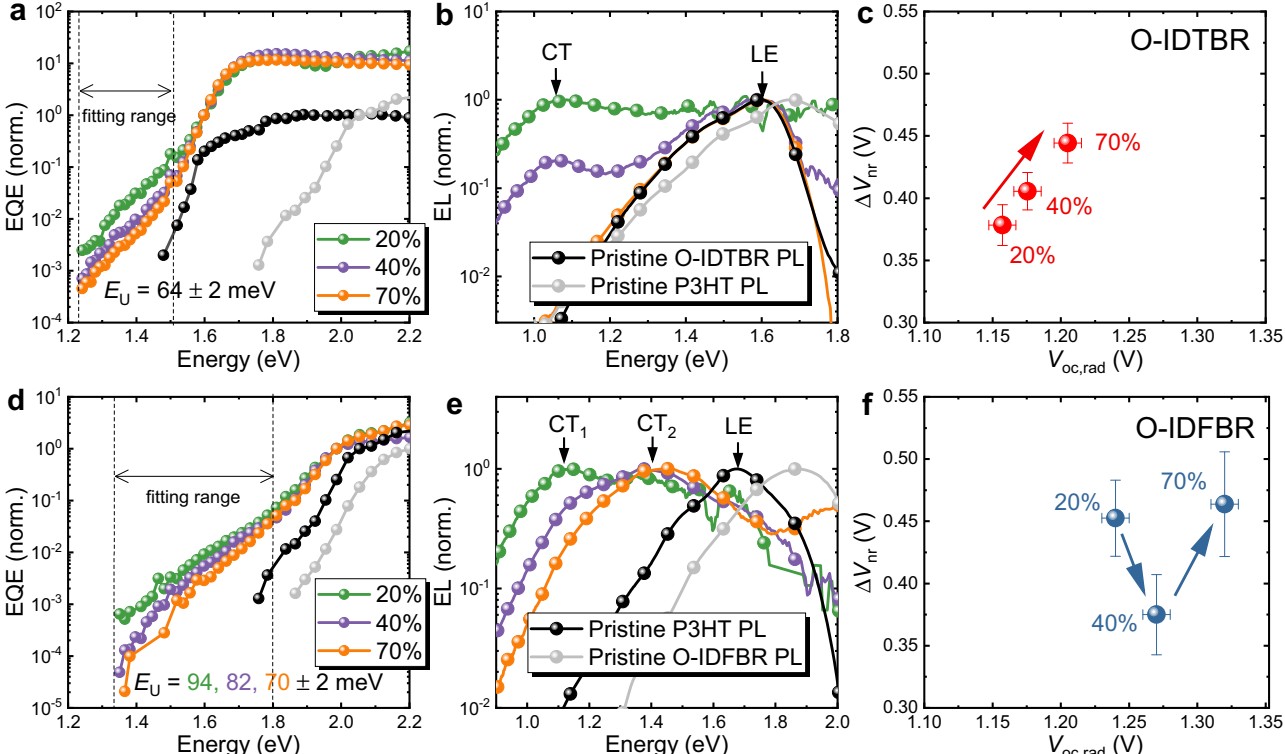

**Fig. 4 Experimental EQE, low injection EL and $V_{loss}$.** Normalized EQE of (**a**) O-IDTBR and (**d**) O-IDFBR devices; Normalized EL of (**b**) O-IDTBR and (**e**) O-IDFBR devices; and $\Delta V_{nr}(V_{oc,rad})$ plot for (**c**) O-IDTBR and **f** O-IDFBR devices. wt% of 20%, 40% and 70% are chosen as representative devices. Urbach energy $E_U$ and the range of fittings are displayed in (**a** and **d**). EQE and photoluminescence (PL) spectra of pristine materials are indicated as black or grey lines. The emission from LE state in the O-IDTBR and O-IDFBR blends are indicated by the PL from pristine O-IDTBR and P3HT (the lower band-gap component), respectively. Note here that the injection current for EL experiments is chosen to be low enough to ensure the devices are close to QTE and to be the same for different NFA wt%. EQE due to CT states is evident in the tail of the blend spectra and can be clearly distinguished from the EQE of the pristine material that is due the absorption of LE states. This means the absorption tail can be assigned to the absorption of CT states in both O-IDTBR and O-IDFBR cases. The error bars in (**c** and **f**) indicate the standard derivation.

the associated loss of interface area and electronic coupling[8,26]; emission from the O-IDTBR exciton dominates the spectrum in the 40% and 70% devices and therefore we cannot clearly observe CT2 in the 40% device or either CT manifold in the 70% device. However, based on our previous study showing that the relative volume fraction of semi-crystalline P3HT stays roughly unchanged at ~70% over compositions from 20% to 70% O-IDTBR[63], we expect that the position and relative density of CT2 remains unchanged with varied composition in the blends of P3HT:O-IDTBR.

Now, we turn to the O-IDFBR blends. Due to the relatively amorphous nature of O-IDFBR, we propose that the two emission peaks in the EL experiments come from amorphous O-IDFBR: semi-crystalline P3HT (CT1) and amorphous O-IDFBR: amorphous P3HT (CT2). Here, as in the case of the P3HT:O-IDTBR blend, the energetic spacing of the two CT states is similar to the energy shift in absorption onset between crystalline and amorphous P3HT[66]. In the 20% O-IDFBR blend, upon increasing injection current, the emission from CT1 reduces while emission from the CT2 manifold increases, as does the relative LE emission. Upon adding more O-IDFBR, we see the relative intensity of emission from CT1 reduce as compared to the CT2 manifold, as seen in Fig. 5e, f. In the 70% O-IDFBR device, emission from CT1 manifold disappears, leaving only emission from CT2. The loss of intensity of CT1 can be rationalized by the disruption of P3HT crystals upon further O-IDFBR addition such that the fraction of amorphous O-IDFBR: semi-crystalline P3HT (CT1) declines[63]. We can also see a clear peak shift of CT2 to

higher energy with increased injection current, indicating an amorphous nature of CT2. Somewhat counterintuitively, increasing the amorphous content is thus accompanied by a sharpening of the CT-DoS tail in this system.

**Voltage loss analysis.** Using EQE and EL under low injection, we can then quantify the voltage loss by calculating $V_{oc,rad}$ and $\Delta V_{nr}$ using the method presented in Supplementary Method 2 (or ref. [58]). We note that our method to experimentally quantify $\Delta V_{nr}$ (see Supplementary Method 2) remains valid given the principle of detailed balance is fulfilled[53] with or without static disorder. Different trends as a function of composition can be seen in Fig. 4c, f . In both cases, we see increased $V_{oc,rad}$ with increasing NFA content, which can be explained by the reduced absorption by the CT-state leading to less radiative recombination (Fig. 4a, d)[58]. The resulting $\Delta V_{nr}$ increases with $V_{oc,rad}$ upon increasing O-IDTBR wt%. (Fig. 4c), whereas $\Delta V_{nr}$ reaches a minimum at 40% (wt%) NFA for the O-IDFBR devices (Fig. 4f). The increase of $\Delta V_{nr}$ as a function of O-IDTBR wt% can be rationalized by the reduced emission intensity, as observed in the EL spectrum (Fig. 4b). According to Supplementary Equation (3), less emission would lead to lower $EQE_{EL}$ and therefore higher $\Delta V_{nr}$[53]. For O-IDFBR devices, the reduction of $\Delta V_{nr}$ can be explained by the reduced contribution of non-radiative recombination from lower energy CT states following the energy gap law[7], as can be seen from the EL in Fig. 4e. We finally note that the large $\Delta V_{nr}$ in 70% O-IDFBR device suggests a different mechanism may be involved for this particular blend, which

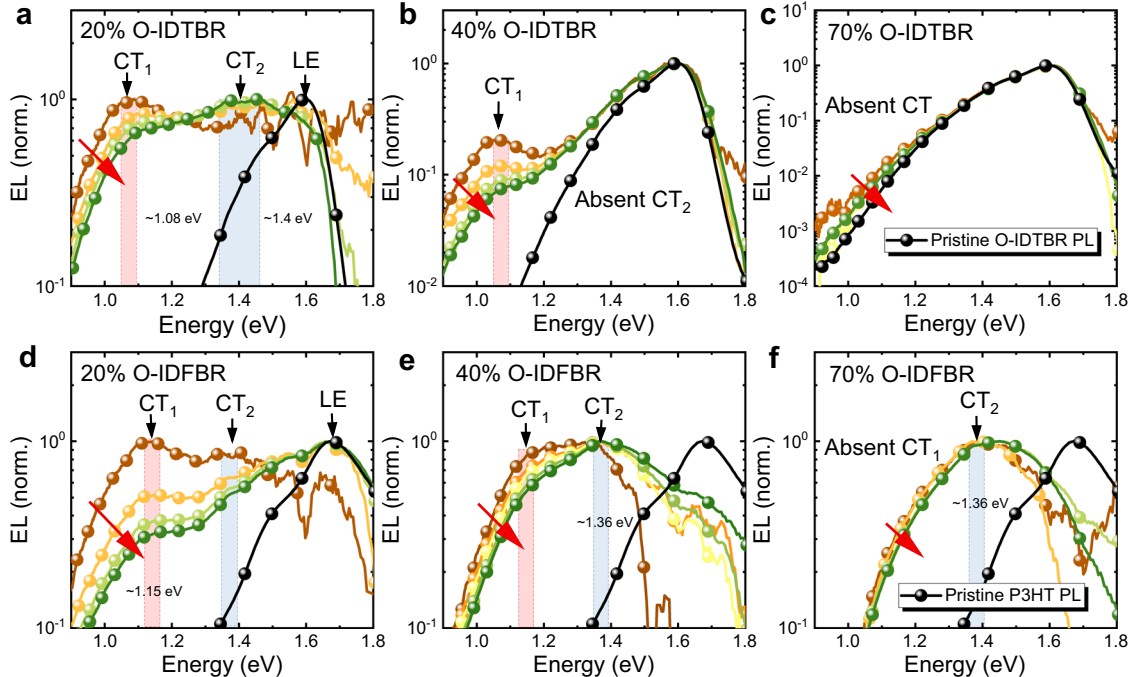

**Fig. 5 Experimental EL with different injection current.** Normalized EL spectrum of O-IDTBR and O-IDFBR devices with (**a**, **d**) 20% wt%, **b**, **e** 40% wt%, and (**c**, **f**) 70% wt% as a function of injection current for EL experiments. Red arrows are indication of increasing injection current. We start with low injection, then slowly increase the injection current to probe the emission from higher energy state. Note here that the range of injection current for different devices are different depending on the brightness of the device, for example, a range of 5–20 mA is used for 40% O-IDFBR device and a range of 20–50 mA is for 20% O-IDFBR device. Local exitonic state emission (LE) is indicated as pristine PL of the low band-gap component, i.e., O-IDTBR in the O-IDTBR blends, and P3HT in the O-IDFBR blends. Different CT states are indicated by the coloured box as well as the text and black arrows.

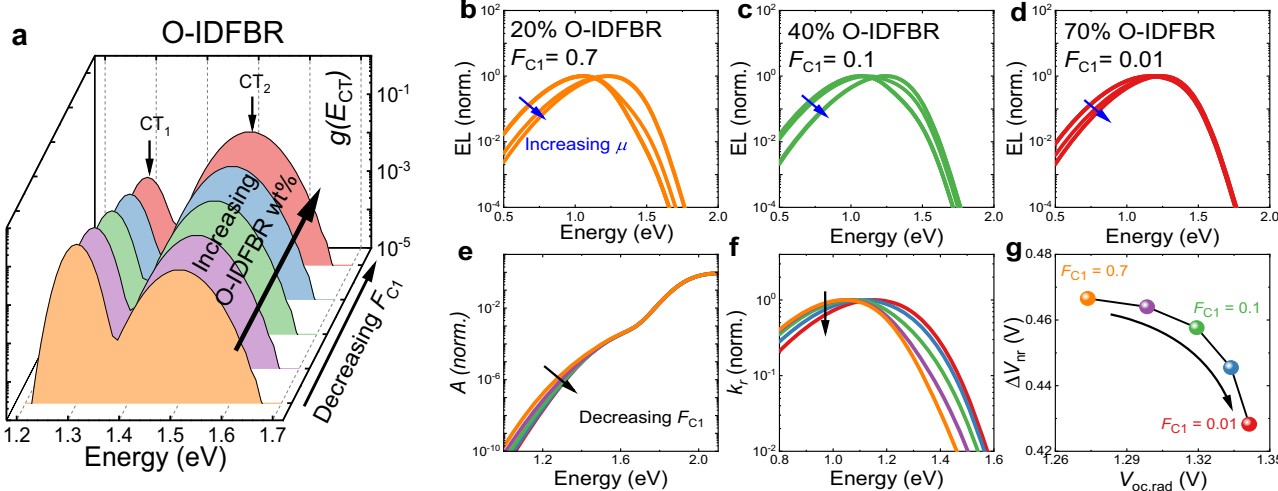

**Fig. 6 Estimated distribution of density of CT state $g(E_{CT})$, and simulated EQE, EL, and $V_{loss}$ for the O-IDFBR devices.** Illustration of the distribution of $g(E_{CT})$ in (**a**) the O-IDFBR devices used in the model. In (**a**) the trend in $g(E_{CT})$ with increasing O-IDFBR fraction from 20 to 70 wt% is modelled via a decrease in the density fraction, $F_{C1}$, of the lower energy CT state from 0.7 to 0.01. $g(E_{CT})$ is modelled using of two CT manifolds with a gaussian width of 0.02 eV (semi-crystalline CT$_1$) and 0.04 eV (amorphous CT$_2$), respectively. The total $g(E_{CT})$ is a superposition of CT$_1$ and CT$_2$. Parts (**b**–**d**) show simulated and normalized injection-dependent emission spectra of O-IDFBR devices with different NFA wt%. Blue arrows are indications of increasing $\mu$ in the range of 1–2 eV. **e**–**g** show absorptance ($A$), rate constant of emission per photon energy ($k_r$), and $\Delta V_{nr}(V_{oc,rad})$ plot. The detailed parameters used to model the O-IDFBR devices are listed in Supplementary Table 6.

cannot be explained by simply considering the reduced contribution from lower energy CT states. The complete device and voltage loss data are presented in Supplementary Figs. 17 and 18.

**Reproducing the experimental trends using the static disorder model.** Our model for the variation in the CT-state distribution

for P3HT:O-IDTBR and P3HT:O-IDFBR devices as a function of composition is illustrated in Supplementary Fig. 19 and summarized in Supplementary Fig. 21a for the O-IDTBR and Fig. 6a the O-IDFBR devices. The emission, absorption and voltage loss behaviour of O-IDTBR devices can be well explained by changes in $f_{osc}$ as a function of composition, without any change in

$g\left(E_{CT}\right)$ (see Supplementary Note 3 for simulation results). Such a loss in CT-state brightness can be expected from the enlargement of the pure donor and acceptor domains[8,26]. However, in the case of O-IDFBR, changes in $g\left(E_{CT}\right)$ upon introducing more O-IDFBR have to be considered in order to rationalize experimental observations as shown below. Such changes or lack of changes in $g\left(E_{CT}\right)$ for P3HT:O-IDFBR and P3HT:O-IDTBR, respectively, are consistent with the previously identified phase behaviours of these blends[63]. We model the $g\left(E_{CT}\right)$ for this blend as two peaks, the lower of which reduces in height, i.e., reducing density fraction of CT$_1$ (notated as $F_{C1}$ in Fig. 6), as O-IDFBR content is increased.

We first model the injection dependence of the EL spectrum (from Fig. 5) on $g\left(E_{CT}\right)$ at different compositions (i.e., O-IDFBR wt%) by varying $\mu$ as described in Supplementary Eq. (24). We note here that if the system obeyed QTE as assumed until now, the EL spectrum would not change shape under increasing injection (see Eq. 11). Since in fact the EL spectrum does change shape upon increasing injection, we need to use a different approach that takes into account the filling of states as previously proposed[52] and as discussed in detail in Supplementary Note 4[9,65]. As shown in Fig. 6b–d, there is a transition in intensity between two peaks with increased $\mu$ in all blend compositions; however, as more O-IDFBR is added (40% w.t. and 70% w.t.), the differences between the peaks begin to be washed out as density fraction of CT$_1$ decreases. For the EQE and low injection EL (Fig. 6e, f), we can see that the tail of absorptance gets sharper and emission from CT$_1$ gets weaker as O-IDFBR wt% increases (Fig. 6e, f), leading to a reduction in $\Delta V_{nr}$. The changes of modelled absorption and emission result from a reducing density fraction of lower energy CT states (i.e., CT$_1$) in Fig. 6a, and the reduction of $\Delta V_{nr}$ results from the reduced non-radiative recombination from CT$_1$ according to the energy gap law[7,44].

These modelling results from Fig. 6 and Supplementary Fig. 19 thus reproduce the key features of the experiments in Figs. 4 and 5, which cannot be reproduced by a single CT-state model for either material system. The resulting $\Delta V_{nr}(V_{oc,rad})$ plots, Supplementary Fig. 21g can explain the O-IDTBR dependence of voltage losses in that blend while Fig. 6g can explain the change from 20% (wt%) to 40% (wt%) of O-IDFBR, although that it fails to rationalize the change from the 40% (wt%) to the 70% (wt%) O-IDFBR device. We note here that at high O-IDFBR content we enter a regime where hole collection is disrupted[63] and that could lead to additional recombination losses due to enhanced non-geminate recombination as well as to an increased rate constant of CT-state decay. However, including this effect is beyond the scope of our model.

We have also explored the impact of changing the energy of the lowest CT-state manifold (CT$_1$) as presented in Supplementary Fig. 22. We note that, in principle, the experimental results could also be fit by an increase in the energy of CT$_1$ with NFA content. The effect of non-equilibrium site distribution on the validity of our analysis is also commented upon in Supplementary Note 5.

## Discussion

This work presents a model that can be used to quantify and understand the impact of disorder in CT state energy (static disorder) on voltage losses in OPV devices. We demonstrated that static disorder tends to increase voltage losses in OPV devices, as previously reported, and that it is indeed important in certain well-studied polymer: molecular acceptor material systems, in agreement with ref. 9,11,17. In the literature it has been common to analyse EQE and emission data in terms of a single CT state. We find that, when spectra from a system which possesses static disorder are interpreted in terms of a single-state energy, the

analysis yields incorrect values of CT-state energy, the associated reorganization energy and voltage losses. It is therefore important to account for disorder in CT-state energy before deriving any material-related trends or guidelines from analysis of experimental data. Several recent studies reporting the relative insignificance of static disorder in CT-state energies in organic blends focussed on dilute dispersion of donor molecules in a fullerene matrix[14,16,18]. Such material systems have been widely studied precisely because morphological disorder is minimised. In contrast, the conjugated polymer:molecular acceptor blends commonly used for high-performance OPV bring several sources of energetic disorder not found in the dilute small molecule blends, namely, the large conformational—and therefore energetic — phase space of conjugated polymers, the electronic anisotropy in both components and the sizeable volume fractions that permit aggregation of both components. In these more widely studied materials, understanding the relationship between phase behaviour and static disorder is important.

Our study explored the relationship between phase behaviour and static disorder in two polymer:small molecule blends, P3HT:O-IDTBR and P3HT:O-IDFBR, where we modulated the interface properties through structure of the small molecule and blend composition. In both cases, different CT-state features appear that are associated with amorphous and crystalline phases of the materials, demonstrating the impact of structural heterogeneity on CT-state energy disorder[26,29,32,33]. Different material phases within the same system tend to enhance the negative effects of CT-state energy disorder. The P3HT:O-IDFBR experimental system helps to validate a key finding from a theoretical model of static disorder, which is that increasing static disorder is generally detrimental to voltage losses (reducing the radiative open-circuit voltage and increasing the non-radiative voltage loss). Given that a sharp onset to the CT-state DoS would help to minimise static disorder, then a blend of two crystalline components may be preferred. However, the lower energy of states in crystalline domains tends to compromise voltage losses via the energy gap law[7,8]. Amorphous components may reduce non-radiative voltage losses that result from high CT-state energy but will generally bring disorder in CT-state energies and will ultimately be compromised by transport limitations. The combination of a low band-gap, sharp onset, crystalline molecule with a less crystalline polymer that presents a low interfacial energy offset would help to minimise static disorder whilst maintaining electronic connectivity. These features are present in the high-performance PM6:Y6 blend, which shows evidence of a lower degree of static disorder both in the shape of its EQE onset and in its temperature dependence than the blends studied[67,68].

Our study demonstrates the importance of accounting for CT-state energy disorder in analysing the voltage losses in OPV devices. Ultimately, approaching the limits to performance will require such static disorder be minimised. In polymer:molecule blends, this will mean reducing the disorder that arises from the conformational phase space of polymers, from aggregation and from molecular anisotropy. At the same time, good electronic and excitonic transport must be maintained. The task will require more detailed experimental[69,70] and theoretical[71] probes of molecular geometry and packing arrangement at interfaces and its relationship with CT-state energy.

**Reporting summary**. Further information on experimental design is available in the Nature Research Reporting Summary linked to this paper.

## Data availability

All the data supporting the current study are available from the corresponding authors upon reasonable request.

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

## Acknowledgements

J.N., J.Y., E.R., M.A. and F.D.E. thank the European Research Council for support under the European Union's Horizon 2020 research and innovation program (Grant Agreement No. 742708). E.R. is grateful to the Fonds de Recherche du Quebec-Nature et technologies (FRQNT) for a postdoctoral fellowship and acknowledges financial support from the European Cooperation in Science and Technology. M.A. thanks the Engineering and Physical Sciences Research Council (EPSRC) for support via doctoral studentships. F.E. thanks the Engineering and Physical Sciences Research Council (EPSRC) for support via the Post-Doctoral Prize Fellowship. We thank Dr. Andrew Wadsworth from Department of Chemistry at Imperial College London and Prof. Iain McCulloch from KAUST Solar Center (KSC) at King Abdullah University of Science and Technology, for providing materials O-IDTBR and O-IDTBR, and extra support of O-IDFBR by Joel Luke from Department of Physics at Imperial College London. J.Y. thank Beverly Ge for her support.

## Author contributions

J.Y. and J.N. proposed the static disorder model and coordinated the experimental part. J.Y. performed theoretical calculations, JV/EQE/EL/PL measurements, and voltage loss analysis. F.D.E. fabricated OPV devices. M.A., F.D.E., E.R. and J.N. gave critical review on the manuscript. J.N. supervised the work.

## Competing interests

The authors declare no competing interests.
