## [Peer Review File · Nature Communications]

REVIEWER COMMENTS

Reviewer #1 (Remarks to the Author):

In this manuscript a theoretical framework for considering the effect of disorder on the charge transfer (CT) state properties of electron donor:acceptor systems for organic photovoltaics is considered. It is shown that CT absorption and emission spectra depend on vibrational broadening and variations in CT state energy due to the local environment in which the CT state is embedded. The model assumes quasi thermal equilibrium and photon absorption and emission are related by detailed balance. The model is applied to polymer:acceptor blends and used to explain the compositional dependence (donor:acceptor ratio) of the voltage losses.

The manuscript summarizes well the state-of-the-art and motivates in a clear manner the importance of a good understanding of the CT state properties. Sufficient details to reproduce the model and simulated curves are given. Overall, this is high-quality work which should be published. I have a few comments for the authors consideration:

- The first sentence of the introduction defines a charge-transfer state. The IUPAC definition of a charge transfer state does not fully correspond to the definition given:

<https://goldbook.iupac.org/terms/view/C01006>, which makes no statements on the Coulomb binding energy. Whether or not positive and negative charge carrier in the CT state are Coulombically bound is irrelevant for the remainder of the paper.

- non-radiative voltage losses are typically calculated in several ways, among them (i) using a single CT state analysis, (ii) from a comparison between the measured open-circuit voltage and the radiative limit of the open-circuit voltage or (iii) via $kT/e \ln(EQE_EL)$. When authors state that single state analysis overestimates the non-radiative voltage losses, it is not so clear for the reader which methods are still valid. Is $kT/e \ln(EQE_EL)$ still a good method to determine non-radiative voltage losses? Can one say something about the validity of the single state model for systems where the single state analysis gives the same non-radiative voltage losses as $kT/e \ln(EQEEL)$?

Reviewer #2 (Remarks to the Author):

This MS reports simulations with an extended model of static disorder in the charge-transfer (CT) states of polymer donor:NFA blends. It analyses the impact of disorder on decay rates of CT states that have been associated previously to voltage losses in OPV cells. This connection is an active research topic in these days.

Going beyond a single gaussian function in the density of CT states, the authors use a series of gaussians in the model. The influence of this extension on radiative and non-radiative rates are studied for two systems. In addition, they perform experiments for P3HT:NFA blends with two different NFAs.

The report is an assembly of simulations with the model and of additional experimental data on the P3HT systems. Each part contains interesting aspects; however, I find the MS not well organised. In part, its presentation is superficial and overall it lacks a clear focus. When extended, it might be more suitable for specialised journals on the topic, given that the large number of issues below be addressed satisfactorily.

The points below should be useful for the authors to rewrite their MS.

1) In the introduction the authors write "To our knowledge no studies to date have explicitly linked static disorder to voltage losses in OPV." In view of the known connection between disorder, spectral linewidth and voltage losses, the statement has to be revised.

2) For the first part in the results section (i.e. eq 2) it is not clear how does it connect to the extension

of the model? In particular, it is not mentioned where this expression comes from?

3) In the theory section, please state the general theory framework used. The approximations behind the rate formulas are not explained.

4) How does the BO approximation enter the model?

5) How is the BO approximation related to the use of the FC factors. Both describe usually the opposite limit.

6) Definition of EQE_{el} integral in supplemental material contains a typo.

7) What is the “chemical potential of the system”?

8) The quantity W is not defined.

9) What are assumptions behind the model for the oscillator strength?

10) What is q in this expression? Why does M depend on omega?

11) Reference or derivation of Eq. S5 should be provided.

12) It seems that the simulations in Fig. 2 are still based on a single Gaussian disorder model, is this true? If so, it is confusing to start with the “old model” and a clearer way of presentation is required.

13) What do the authors mean by saying they check whether their “results obey the principle of detailed balance”? From the text, I understood that they enforce this in the model. Clarification and explanation is required.

14) I don't see why the validity of their model depends on the principle of detailed balance? The assumptions underlying the model should be stated more clearly.

15) It seems that Fig. 2a and 2b show the same results. Why are they needed both?

16) Same data is labeled kr in one subfigure and kr/Phi_{BB} in the other.

17) Figure 2d is not mentioned in the text.

18) As a general remark, I find the discussion of the results not suitable for the general and the expert reader, because it is not explained in an accessible way. This makes it extremely hard for the reader to grasp the significance of the work. The discussion of the results is very superficial. Sometimes the formulas that are used to obtain the results are not mentioned. Sometimes only one sentence is written and the authors proceed to the next part.

19) What do we learn from Figure 2e? Why the comparison to Ref. 7?

20) Figure 4 shows EQE spectra, whose edges are fitted to extract Urbach energies. How can the authors be sure this part is from CT states? Nothing has been mentioned to support this assumption. I am wondering about the trend with increasing wt% in both cases Fig. 4a and Fig. 4d. Why does the CT signal decrease with higher NFA content?

21) The authors just say that the result was interesting and state the numbers for the Urbach energies for the blends. But what does the EQE data contribute to the MS and the understanding of these systems?

22) Figure 4c and 4f are not discussed beyond simply stating what the reader can see anyhow in the figure.

23) A strong assumption in the paper is about the origin of the emission peaks at low energies (Fig. 5). This is the basis for parts of the discussions. The authors state “We suggest that the two peaks are induced by interfaces with semi-crystalline polymer (lower energy CT1) and interfaces with amorphous polymer (higher energy CT2)”. This seems to remain a difficult point to verify, despite the EL data at hand. Is there additional evidence that can be used to consolidate this assumption?

24) Why does CT2 disappear in Fig. 5b with decreasing P3HT wt%?

25) The authors continue with a two-state CT model (Fig. 6) in which additional parameters enter to fit the experimental data. The connection of the model parameters such as modification in f_{osc} or relative weights of CT1 and CT2 should be better motivated and explained. This is not very transparent. How are F_{C1} and F_{C2} defined?

26) Fig. 6a and 6b are not clear about the parameter on the y-axis, is this the NFA content (which range)? Additional assumptions should be made clearer in a different way.

27) In the models for Fig. 6, is there any LE state? Nothing has been mentioned or explained in the theory section.

28) Please clarify how the specified voltage enters.

29) In Fig. 6, the differences in the results between Fig. 6a and Fig. 6b and the connection to the differences in the modelling of both NFA compounds is not discussed. Only the curves are described.

30) I find that the “agreement with experiment in Fig. 5” cannot be concluded in this way. From comparing the figures, the differences are obvious.

31) The last paragraph on page 14 basically claims qualitative agreement without saying what is meant by this. This is a low level of presentation and needs to be improved. Possibly this is due to the large number of results included in the MS and limited space. Therefore a more specialised journal is more suitable.

32) Coming to the summary, the authors claim "This work presents a model that can be used to quantify and understand the impact of disorder in charge-transfer state energy (static disorder) on voltage losses in organic photovoltaic devices." Given that the discussion of many aspects of their work is unfortunately very superficial and that in particular the discussion of voltage losses is basically absent from the MS, this claim cannot be sustained.

33) As mentioned in my initial statements, in my opinion this MS suffers from a lack of a clear focus. The statements in the abstract/conclusions are only partly aligned with the text in the MS. It seems that the choice of experimental studies is not so ideal since the P3HT:NFA blends exhibit complicated (and only partly known) phase behavior and quite specific CT state features. At the same time, both systems are also rather similar to each other. In my opinion more space is needed to properly discuss both systems.

34) The relevance of the generality of the results is obscured by the specific properties of the selected systems. It is unclear why both are needed in order to arrive at most of the conclusions presented, such as "We demonstrated that static disorder tends to increase voltage losses in OPV devices, as previously reported, and that it is indeed important in certain well-studied polymer:molecular acceptor material systems, in agreement with Ref. 9,11,17."

35) The sentence "It is therefore important to account for disorder in CT state energy before deriving any material related trends or design rules from analysis of experimental data." appears somewhat unrelated to the MS because it does not aim at any design rules.

36) It is not clear what is meant by the conclusion "Our combined experimental and modelling study demonstrates how the model can be used to explore the relationship between phase behaviour and static disorder in a blend.", because the relationship between phase behaviour (not studied in the MS) and static disorder has not been studied but has been suggested by the authors.

37) The phrase "experimental validation of the model" in the abstract is questionable in view of the specific input needed to reproduce the experiments. It appears that the model has been tailored to reproduce some experiments.

Reviewer #3 (Remarks to the Author):

The authors of this manuscript suggest that static disorder is the dominant contribution and present a physical model to calculate the influence of disorder on voltage (and, thus, efficiency) losses in OPVs. This model is applied to two prototypical polymer:non-fullerene blends showing different impact of their mixing ratio on the interfacial donor-acceptor morphology and, thus, the CT DOS. Model predictions and experimental results are compared to obtain valuable insights in voltage losses of these devices.

Overall, the manuscript is sound and represents a significant step forward to better understanding OPV devices. It can be published after some issues have been addressed:

1. Before presenting their own model, the authors list previous work and the related shortcomings in the context of properly including static disorder effects (p. 4). Specifically, they pick the model by Kahle et al. (Ref. 11) to demonstrate that it does not fulfil detailed balance (Fig. S1). However, it is not specified which of the different expressions given in Ref. 11 is actually used for the simulation; let alone the required input parameter values to produce the curves shown in Fig. S1. Moreover, the authors should add some discussion in the SI as to why this expression does not fulfil DB. I think it should even be possible to show this analytically (in addition to the simulation).

2. The new model is based in part on work published earlier in 2018 (Ref. 8). Nevertheless, the authors should try to present the theory in a way that readers get the basic idea without having to dig too deep in the literature. This holds especially for the presentation on p. 6 of the manuscript. The expressions for the FCWD – by the way, the acronym needs to be defined as well – appear all by a sudden without any introduction how they are to be used. I think it would be much easier to follow if

parts of section 3 from the SI are transferred to the main manuscript text.

3. Furthermore, I find it hard to compare simulation results if the input parameters are not fully specified (see e.g. Fig. 3 or Fig. S3). Often, only the varied parameter values are given (Fig. S4-S8) but the rest is unclear. I suggest to list all input parameter, e.g. in another table in the SI.

4. The application of the model to experimental data is performed in a qualitative manner without fitting. Nevertheless, it seems to yield valid results. Just two comments: (i) While temperature dependent EQE indicates that static disorder is dominant (Fig. S15), there is no information about T-dependent EL spectra. (ii) Is it possible to label the energy axis of Fig. S18?

5. There are some typos that should be corrected:

- Fig. 1, caption: "triple" should read "triplet"
- Fig. 6, caption: Where in the graphs do I find "F_C1" and "F_C2"?
- SI, section 1: Lockin is probably an SRS830 (and not 380) model. Amplification probably 10^6 V/A (and not 106)!
- Fig. S11-13: Legend should read "Dashed".
- Fig. S11: Caption should refer to Fig. 3a (and not 2a) of main text.

Point-by-point Response to the Reviewers' Comments

Manuscript NCOMMS-20-48130

We thank the reviewers for their comments and constructive suggestions. Please find below the point-by-point responses to the comments. To ease readability, we are repeating the reviewers' comments in *italics*.

Reviewer(s)' Comments to Author:

Report of Reviewer #1:

Reviewer's overall evaluation:

In this manuscript a theoretical framework for considering the effect of disorder on the charge transfer (CT) state properties of electron donor:acceptor systems for organic photovoltaics is considered. It is shown that CT absorption and emission spectra depend on vibrational broadening and variations in CT state energy due to the local environment in which the CT state is embedded. The model assumes quasi thermal equilibrium and photon absorption and emission are related by detailed balance. The model is applied to polymer:acceptor blends and used to explain the compositional dependence (donor:acceptor ratio) of the voltage losses.

The manuscript summarizes well the state-of-the-art and motivates in a clear manner the importance of a good understanding of the CT state properties. Sufficient details to reproduce the model and simulated curves are given. Overall, this is high-quality work which should be published. I have a few comments for the authors consideration:

Responses:

We appreciate very much **Reviewer #1** for her/his positive views on our manuscript. We also sincerely thank the her/him for the following suggestions that led us to further improve the manuscript.

Comment 1:

The first sentence of the introduction defines a charge-transfer state. The IUPAC definition of a charge transfer state does fully correspond to the definition given: <https://goldbook.iupac.org/terms/view/C01006>, which makes no statements on the Coulomb

binding energy. Whether or not positive and negative charge carrier in the CT state are Coulombically bound is irrelevant for the remainder of the paper.

Responses:

Following this comment, we have revised the first sentence in the main text (see also below in blue):

“A charge-transfer (CT) state at a donor-acceptor (D-A) interface in an organic photovoltaic (OPV) device is an intermediate state present after a charge transfer transition in which the electron (on the acceptor) and the hole (on the donor) reside on either side of the interface.”

Comment 2:

non-radiative voltage losses are typically calculated in several ways, among them (i) using a single CT state analysis, (ii) from a comparison between the measured open-circuit voltage and the radiative limit of the open-circuit voltage or (iii) via $kT/e \cdot \ln(EQE_{EL})$. When authors state that single state analysis overestimates the non-radiative voltage losses, it is not so clear for the reader which methods are still valid. Is $kT/e \cdot \ln(EQE_{EL})$ still a good method to determine non-radiative voltage losses? Can one say something about the validity of the single state model for systems where the single state analysis gives the same non-radiative voltage losses as $kT/e \cdot \ln(EQE_{EL})$?

Responses:

We thank the reviewer for this important question.

For Method (i), we have shown in the manuscript theoretically that when the device has a significant amount of static disorder single state analysis can give wrong values of E_{CT} and the outer reorganization energy (λ), and could eventually overestimate ΔV_{nr} . So, Method (i) is problematic when static disorder is present. In the case when ΔV_{nr} calculated from single state analysis equals to the values obtained from method (iii) (i.e. via $\Delta V_{nr} = \frac{k_B T}{q} \ln(EQE_{EL})$), we would expect that the measured device presents a small amount of static disorder as we can see in Supplementary Fig. 11.

Method (ii) and (iii) are essentially based on the reciprocity relation between photon absorption and emission (see Refs. J. Phys. C: Solid State Phys. 15 3967 (1982) and Physical Review B 76 (8), 085303 (2007)). They both are absolute measurements of nonradiative voltage losses, so we believe both methods are valid given the principle of detailed balance is fulfilled in the systems studied.

We have also added some text to discuss the key points in the main text (Line 262) in blue:

“We therefore conclude that the single state analysis is only valid when there is little or no static disorder present (see Supplementary Fig. 11).”

And Line 361 in Section “Voltage loss analysis”:

“We note that our method to experimentally quantify ΔV_{nr} (see Supplementary Method 2) remains valid given the principle of detailed balance is fulfilled⁵³ with or without static disorder.”

Report of Reviewer #2:

Reviewer's overall evaluation:

This MS reports simulations with an extended model of static disorder in the charge-transfer (CT) states of polymer donor:NFA blends. It analyses the impact of disorder on decay rates of CT states that have been associated previously to voltage losses in OPV cells. This connection is an active research topic in these days.

Going beyond a single gaussian function in the density of CT states, the authors use a series of gaussians in the model. The influence of this extension on radiative and non-radiative rates are studied for two systems. In addition, they perform experiments for P3Ht:NFA blends with two different NFAs.

The report is an assembly of simulations with the model and of additional experimental data on the P3HT systems. Each part contains interesting aspects; however, I find the MS not well organised. In part, its presentation is superficial and overall it lacks a clear focus. When extended, it might be more suitable for specialised journals on the topic, given that the large number of issues below be addressed satisfactorily.

The points below should be useful for the authors to rewrite their MS.

Responses:

We appreciate very much that **Reviewer #2** read the manuscript thoroughly and her/his valuable and critical comments. Below we have responded to every comment and made relevant revisions in both the main text and the SI. We also believe the reviewer's comments have strengthened the manuscript.

Comment 1:

1) In the introduction the authors write "To our knowledge no studies to date have explicitly linked static disorder to voltage losses in OPV." In view of the known connection between disorder, spectral linewidth and voltage losses, the statement has to be revised.

Responses:

Thanks very much for pointing this out, and we apologize for the incorrect statement, and we have therefore deleted this sentence in the introduction.

Comment 2-5:

2) For the first part in the results section (i.e. eq 2) it is not clear how does it connect to the extension of the model? In particular, it is not mentioned where this expression comes from?

3) In the theory section, please state the general theory framework used. The approximations behind the rate formulas are not explained.

4) How does the BO approximation enter the model?

5) How is the BO approximation related to the use of the FC factors. Both describe usually the opposite limit.

Responses:

We appreciate the reviewer's request for more details of the model, which were raised in a similar way by **Reviewer #3**. Here we respond to comments 2-5 together, since they are related.

Following the comments, we have added some introduction on the general theory framework (based on Ref. 42-44: J. Am. Chem. Soc. 116, 8188–8199 (1994), Chem. Phys. 176, 439–456 (1993), and J. Chem. Phys. 64, 4860 (1976)), and have added the key approximations and revised the section “Static disorder model” in the main text.

Regarding the use of Born-Oppenheimer (BO) approximation and Franck-Condon (FC) factor, to be clearer, BO approximation is embedded in the derivation of the rate equations, as we need to separate the electronic part from nuclear part of the wavefunction, which is based on BO approximation. The Franck-Condon weighted density then comes in to replace the nuclear part wavefunction along with the delta function in the rate equations, such that the semi-classical Marcus rate equations (also the Marcus–Levich–Jortner (MLJ) rate equations) can be derived. To make it clearer in the text, the relevant parts (“Static disorder model” section in the main text and Supplementary Method 3 in the SI) in the present manuscript have been carefully revised and reviewed by the authors.

Changes have been made to the model section in the main text and Supplementary Method 3 as highlighted in blue (see also below):

“Static disorder model” section in the main text:

“Static disorder model. Fig. 1a-b illustrates a general energetic distribution of CT states, $g(E_{CT})$, that might result at a D-A heterointerface in which a donor domain is surrounded by acceptor domains of different sizes and strength of interaction with the donor²⁸. In general, such a distribution is likely to contain a number of CT-state manifolds, each centred at $E_{CT,ct}$, where t denotes the order of CT manifold. We thus describe the static disorder in the system by a normalised distribution function for CT-state energies given by

$$g(E_{CT}) = \sum_t c_t D_t(E_{CT}), (1)$$

where $D_t(E_{CT})$ is a line-shape function and c_t is a weighting coefficient such that $\sum_t c_t = 1$. Practically, $D_t(E_{CT})$ can be any function. In our model we use a gaussian line-shape for $D_t(E_{CT})$, i.e.

$$D_t(E_{CT}) = \frac{1}{\sigma_{CT,t}\sqrt{2\pi}} \exp\left[-\frac{1}{2}\left(\frac{E_{CT}-E_{CT,ct}}{\sigma_{CT,t}}\right)^2\right],$$

where $\sigma_{CT,t}$ is the width of the individual gaussian function. In the calculations shown below, each gaussian function is integrated in the range of $[E_{CT,ct} - 5\sigma_{CT,t}, E_{CT,ct} + 5\sigma_{CT,t}]$, ensuring $\int_a^b D_t(E_{CT})d(E_{CT}) = 1$, hence $\int g(E_{CT})d(E_{CT}) = 1$, where $a = E_{CT,ct} - 5\sigma_{CT,t}$ and $b = E_{CT,ct} + 5\sigma_{CT,t}$.

As in previous models^{7,8,38}, we assume that radiative and non-radiative recombination in the device follow the energy gap law^{43,44}, and occur only via the CT states, either directly after exciton dissociation or by reformation of the CT state from free charges. We additionally assume that absorption, emission and non-radiative recombination transitions occur in the weak coupling limit so that they can be described by non-adiabatic Marcus theory, and we invoke the Franck Condon principle to include transitions between different vibrational modes, as illustrated in Fig. 1c. Here, we adopt the method introduced by Jortner to describe the rate of transition between the CT

state and the ground state⁴⁵, in which we distinguish between high and low frequency vibronic modes. We consider that the states in each CT state manifold (denoted as t in the subscript) share the same set of parameters, such as oscillator strength ($f_{osc,t}$) and low frequency reorganization energies ($\lambda_{o,t}$), except for their energies (E_{CT}). The total rate of non-radiative recombination from the CT state can be expressed as the sum of the contribution from all CT manifolds, as:

$$K_{nr} = \sum_t \int_a^b \frac{2\pi}{\hbar} V_t(E_{CT})^2 FCWD_{rec,t}(0, E_{CT}) c_t D_t(E_{CT}) d(E_{CT}), \quad (2)$$

where $V_t(E_{CT})$ is the electronic coupling between CT and ground state described by the generalized Mulliken-Hush method^{46,47}. $FCWD_{rec,t}$ is the Franck-Condon-Weighted Density of States for recombination for CT state manifold t . The radiative recombination and absorption rates can be expressed as the sum of the contribution from all CT manifolds using a similar expression and all depend on the Franck-Condon-Weighted Density of States (FCWD)⁸, via

$$k_{abs}(\hbar\omega) = \sum_t \int_a^b \frac{W}{3\pi\epsilon_0\hbar^4} \left(\frac{\hbar\omega}{c}\right)^3 M_t(E_{CT})^2 FCWD_{abs,t}(\hbar\omega, E_{CT}) c_t D_t(E_{CT}) d(E_{CT}), \quad (3a)$$

$$k_r(\hbar\omega) = \sum_t \int_a^b \frac{1}{3\pi\epsilon_0\hbar^4} \left(\frac{\hbar\omega}{c}\right)^3 M_t(E_{CT})^2 FCWD_{rec,t}(\hbar\omega, E_{CT}) c_t D_t(E_{CT}) d(E_{CT}), \quad (3b)$$

where W is the photon density and accounts for the strength of electromagnetic field around the molecule;⁴⁸ ϵ_0 is the permittivity of the free space; $M_t(E_{CT})$ is the transition dipole moment for CT manifold t and is related to the oscillator strength of the CT manifold ($f_{osc,t}$) under the dipole approximation^{49,50} via $M_t(E_{CT}) = \sqrt{(3/2)q^2\hbar^2 f_{osc,t}/(E_{CT}m_e)}$, with m_e the electron mass and q the elementary charge. K_r (s^{-1}) is obtained by integrating Equation (3b) over $\hbar\omega$, via

$$K_r = \int_0^{\infty} k_r(\hbar\omega) d\hbar\omega. (4)$$

We assume quasi-thermal equilibrium (QTE) conditions, meaning that the occupation function of each electronic CT state should be considered in the expression for photon emission, and that state occupation should follow Boltzmann statistics under low injection conditions. This leads to modified expressions from MLJ theory^{7,8} for FCWD for absorption ($FCWD_{abs,t}(\hbar\omega, E_{CT})$) and recombination ($FCWD_{rec,t}(\hbar\omega, E_{CT})$) for CT state manifold (t),

$$\begin{aligned} & FCWD_{abs,t}(\hbar\omega, E_{CT}) \\ &= \frac{1}{Z_{abs}} \frac{1}{\sqrt{4\pi\lambda_{o,t}k_B T}} \\ &\times \sum_{m=0}^{\infty} \sum_{n=0}^{\infty} \frac{e^{-S_t} S_t^{n-m} m!}{n!} [(L_m^{n-m}(S_t))]^2 \exp\left[-\frac{(-\hbar\omega + E_{CT} + \lambda_{o,t} + (n-m)\hbar\Omega_t)^2}{4\lambda_{o,t}k_B T}\right] \exp\left(-\frac{m\hbar\Omega_t}{k_B T}\right), (5a) \end{aligned}$$

$$\begin{aligned} & FCWD_{rec,t}(\hbar\omega, E_{CT}) \\ &= \frac{1}{Z_{rec}} \frac{1}{\sqrt{4\pi\lambda_{o,t}k_B T}} \\ &\times \sum_{m=0}^{\infty} \sum_{n=0}^{\infty} \frac{e^{-S_t} S_t^{n-m} m!}{n!} [(L_m^{n-m}(S_t))]^2 \exp\left[-\frac{(\hbar\omega - E_{CT} + \lambda_{o,t} + (n-m)\hbar\Omega_t)^2}{4\lambda_{o,t}k_B T}\right] \exp\left(-\frac{m\hbar\Omega_t + E_{CT} - \mu}{k_B T}\right), (5b) \end{aligned}$$

where $\lambda_{o,t}$ and $\lambda_{i,t}$ are the low frequency and high frequency reorganization energy for CT manifold t , respectively, $S_t = \lambda_{i,t}/\hbar\Omega_t$ is the Huang Rhys factor⁵¹, $\hbar\Omega_t$ is the averaged harmonic energy spacing, typically 0.15-0.20 eV for molecules made of many carbon-carbon bonds^{7,8} and m and n are the quantum numbers of the vibrational mode of the initial and final state, respectively. $L_m^{n-m}(S_t)$ is then the generalized Laguerre polynomial of degree m ⁸. The constraint of equilibrated occupation of electronic states following Boltzmann statistics ensures that the different D-A interfaces in the ensemble share the same chemical potential energy (μ) and the system is in QTE. μ is defined as the free energy gained by the system under injection or illumination conditions, and is also known as the Fermi energy in

inorganic semiconductors. The factor Z_{abs} and Z_{rec} in Equation (5) are the partition functions, and are defined as the sum of occupation of vibrational CT states for absorption, and as the sum of occupation of both vibrational and electronic CT states for recombination, and are given by

$$Z_{abs} = \sum_t \sum_{m=0}^{\infty} \exp\left(-\frac{m\hbar\Omega_t}{k_B T}\right), \quad (6a)$$

$$Z_{rec} = \sum_t \left\{ \sum_{m=0}^{\infty} \exp\left(-\frac{m\hbar\Omega_t}{k_B T}\right) \int_a^b c_t D_t(E_{CT}) \exp\left(-\frac{E_{CT} - \mu}{k_B T}\right) d(E_{CT}) \right\}. \quad (6b)$$

When the absorption and emission spectra are constructed from these FCWD distributions, the spectra represent the sum of contributions from different CT state energies, and the resulting absorption and emission bands are broadened not only by the low frequency broadening of each vibronic line and the FC vibronic progression, but also by the distribution of CT state energies. Attempts to interpret such spectra in terms of a single CT state energy would lead to incorrect estimates for both the CT state energy and the reorganisation energies. We also remark in passing that in the case where the whole system is not in QTE, for example if the spatially separated states are not strongly coupled, as explored by Melianas et al⁵², we would, in general, expect different chemical potential energies at different manifolds (sites) of CT states, and the above expression for emission would need to be adapted. The full model describing the voltage losses as well as the rates of the radiative and non-radiative transitions with static disorder included is presented in Supplementary Method 2-4.”

Supplementary Method 3 in the SI:

“To express the radiative recombination rate, we consider the rate of emission from the CT state to the ground state. We express the radiative rate constant using the operator ($O = \frac{e}{mc} \vec{A} \cdot \vec{p}$), \vec{A} is the vector potential of the electromagnetic field and \vec{p} the momentum operator⁶.

With static disorder involved, using the Fermi's Golden (FG) rule and the dipole approximation, the emission rate (k_r) per unit photon energy ($\hbar\omega$) can be expressed using the transition dipole moment.⁸ The absorption rate per photon energy (k_{abs}) can be expressed in a similar way as k_r except that we need to consider the photon density (W) in the volume. We consider the states in each CT state manifold (denoted as i in the subscript) share the same set of parameters. Ultimately, when summing over all contributions from different CT manifolds (denoted as t), we have

$$k_{abs}(\hbar\omega) = \sum_t \int_a^b \frac{W}{3\pi\epsilon_0\hbar^4} \left(\frac{\hbar\omega}{c}\right)^3 M_t(E_{CT})^2 FCWD_{abs,t}(\hbar\omega, E_{CT}) c_t D_t(E_{CT}) d(E_{CT}), \quad (M3.1a)$$

$$k_r(\hbar\omega) = \sum_t \int_a^b \frac{1}{3\pi\epsilon_0\hbar^4} \left(\frac{\hbar\omega}{c}\right)^3 M_t(E_{CT})^2 FCWD_{rec,t}(\hbar\omega, E_{CT}) c_t D_t(E_{CT}) d(E_{CT}), \quad (M3.1b)$$

Where W is the photon density and accounts for the strength of electromagnetic field around the molecule;⁹ ϵ_0 is the permittivity of the free space; t is the order of CT state manifold; $M_t(E_{CT})$ is the transition dipole moment for CT manifold t and is related to the oscillator strength of the CT manifold ($f_{osc,t}$) under the dipole approximation^{10,11}, via $M_t(E_{CT}) = \sqrt{(3/2)q^2\hbar^2 f_{osc,t}/(E_{CT}m_e)}$, with m_e the electron mass. $FCWD_{abs,t \text{ or } rec,t}(\hbar\omega, E_{CT})$ is the Franck-Condon weighted density of state for absorption or emission at $\hbar\omega$ for a certain CT manifold t , and it sums all the wavefunction overlaps between the vibrational modes of initial and final state, as follow

$$FCWD_{abs,t}(\hbar\omega, E_{CT}) = \frac{1}{Z_{abs}} \frac{1}{\sqrt{4\pi\lambda_{o,t}k_B T}} \times \sum_{m=0}^{\infty} \sum_{n=0}^{\infty} \frac{e^{-S_t} S_t^{n-m} m!}{n!} [L_m^{n-m}(S_t)]^2 \exp\left[-\frac{(-\hbar\omega + E_{CT} + \lambda_{o,t} + (n-m)\hbar\Omega_t)^2}{4\lambda_{o,t}k_B T}\right] \exp\left(-\frac{m\hbar\Omega_t}{k_B T}\right), \quad (M3.2a)$$

$$\begin{aligned}
& FCWD_{rec,t}(\hbar\omega, E_{CT}) \\
&= \frac{1}{Z_{rec}} \frac{1}{\sqrt{4\pi\lambda_{o,t}k_B T}} \\
&\times \sum_{m=0}^{\infty} \sum_{n=0}^{\infty} \frac{e^{-S_t} S_t^{n-m} m!}{n!} [(L_m^{n-m}(S_t))^2] \exp\left[-\frac{(\hbar\omega - E_{CT} + \lambda_{o,t} + (n-m)\hbar\Omega_t)^2}{4\lambda_{o,t}k_B T}\right] \exp\left(-\frac{m\hbar\Omega_t + E_{CT} - \mu}{k_B T}\right), \quad (M3.2b)
\end{aligned}$$

where $\lambda_{o,t}$ and $\lambda_{i,t}$ are the low frequency and high frequency reorganization energy for CT manifold t , respectively, $S_t = \lambda_{i,t} / \hbar\Omega_t$ is the Huang Rhys factor¹², $\hbar\Omega_t$ is the averaged harmonic energy spacing, typically 0.15-0.20 eV for molecules made of many carbon-carbon bonds^{6,13} and m and n are the quantum numbers of the vibrational mode of the initial and final state, respectively. $L_m^{n-m}(S_t)$ is then the generalized Laguerre polynomial of degree m ⁶. The constraint of equilibrated occupation of electronic states following Boltzmann statistics ensures that the different D-A interfaces in the ensemble share the same chemical potential energy (μ) and the system is in QTE. μ is the free energy gained by the system under injection or illumination conditions, and is also known as Fermi energy in semiconductors. The factor Z_{abs} and Z_{rec} in Equation (M3.2) are the partition functions, and are defined as the sum of occupation of vibrational CT states for absorption, and as the sum of occupation of both vibrational and electronic CT states for recombination, and are given by

$$Z_{abs} = \sum_t \sum_{m=0}^{\infty} \exp\left(-\frac{m\hbar\Omega_t}{k_B T}\right), \quad (M3.3a)$$

$$Z_{rec} = \sum_t \left\{ \sum_{m=0}^{\infty} \exp\left(-\frac{m\hbar\Omega_t}{k_B T}\right) \int_a^b c_t D_t(E_{CT}) \exp\left(-\frac{E_{CT} - \mu}{k_B T}\right) d(E_{CT}) \right\}. \quad (M3.3b)$$

K_r (s^{-1}) is obtained by integrating Supplementary Equation (M3.1b) over $\hbar\omega$, and K_{nr} (s^{-1}) can be expressed in a similar fashion as k_r (see Supplementary Equation (M3.1b)) using Fermi's Golden (FG) rule¹⁴ and BO approximation¹⁵ as a product of the electronic coupling V between the CT state and ground state and the Franck-Condon weighted density of states. And we have

$$K_r = \int_0^{\infty} k_r(\hbar\omega) d\hbar\omega, \text{ (M3.4a)}$$

$$K_{nr} = \sum_t \int_a^b \frac{2\pi}{\hbar} V_t(E_{CT})^2 FCWD_{rec,t}(0, E_{CT}) c_t D_t(E_{CT}) d(E_{CT}), \text{ (M3.4b)}$$

where $V_t(E_{CT})$ is the electronic coupling (EC) between CT and ground state described by generalized Mulliken-Hush method^{16,17}. $FCWD_{rec,t}(0, E_{CT})$ follows Supplementary Equation (M3.2b) with $\hbar\omega = 0$.

We have now the rate constants K_r and K_{nr} to model ΔV_{nr} using Supplementary Equation (M2.3). To predict V_{oc} , what we also need is $V_{oc,rad}$, which is calculated using $EQE_{abs}(\hbar\omega)$ in Supplementary Equation (M2.2).”

Comment 6:

6) Definition of EQE_{el} integral in supplemental material contains a typo.

Responses:

Following this comment, we have simplified and corrected that sentence in the SI (see also below in blue):

ΔV_{nr} is determined by the external quantum efficiency of light emission (EQE_{EL}), via

$$\Delta V_{nr} = \frac{k_B T}{q} \ln \left(\frac{1}{EQE_{EL}} \right), \text{ (M2.3)}$$

Comment 7:

7) What is the “chemical potential of the system”?

Responses:

Thank you for raising this question. To make it clearer in the main text, we have revised the paragraph in page 7 (Line 147) (see below the changes in blue):

“The constraint of equilibrated occupation of electronic states following Boltzmann statistics ensures that the different D-A interfaces in the ensemble share the same chemical potential energy (μ) and the system is in QTE. μ is defined as the free energy gained by the system under injection or illumination conditions, and is also known as the Fermi energy in inorganic semiconductors.”

Comment 8:

8) *The quantity W is not defined.*

Responses:

We apologize for this and have defined the quantity W in the main text (line 126 in page 6) (see also below in blue).

“Where W is the photon density and accounts for the strength of electro-magnetic field around the molecule;”

Comment 9 and 10:

9) *What are assumptions behind the model for the oscillator strength?*

10) *What is q in this expression? Why does M depend on ω ?*

Responses:

We agree the two questions need clarifying, here we respond them together as they are related.

The assumption behind the oscillator strength is the dipole approximation, therefore, the oscillator (f_{osc}) can be expressed as a function of the transition dipole moment (M) (see Fox, M. Optical Properties of Solids (Oxford Univ. Press, 2001)):

$$f_{osc}(E_{CT}) = (2/3)M(E_{CT})^2 E_{CT} m_e / (q^2 \hbar^2)$$

And in our model, as we use f_{osc} as the input parameter, we express the transition dipole moment $M(E_{CT})$ as a function of f_{osc} , therefore:

$$M(E_{CT}) = \sqrt{(3/2)q^2\hbar^2 f_{osc}/(E_{CT}m_e)}$$

And q is the elementary charge in the model, and $M(E_{CT})$ is not a function of $\hbar\omega$. Both have been corrected in the manuscript (line 127 in page 6) (see also below in blue):

“ $M_t(E_{CT})$ is the transition dipole moment for CT manifold t and is related to the oscillator strength of the CT manifold ($f_{osc,t}$) under the dipole approximation^{49,50} via $M_t(E_{CT}) = \sqrt{(3/2)q^2\hbar^2 f_{osc,t}/(E_{CT}m_e)}$, with m_e the electron mass and q the elementary charge.”

Comment 11:

11) Reference or derivation of Equation S5 should be provided.

Responses:

Thanks for the suggestion. In correcting this text, we also noticed a typo in this equation: $k_{abs}(\hbar\omega)$ should be replaced by $\alpha_{abs}^{CT}(\hbar\omega)$. We have also provided the reference (Phys. Rev. B 76, 085303 (2007)) and a detailed derivation for the detailed balance between $\alpha_{abs}^{CT}(\hbar\omega)$ and $k_r(\hbar\omega)$. And Supplementary Method 5 in the SI has been revised as follow:

Following the derivation by Rau, to obey the principle of detailed balance, the following equations should be validated²:

$$A(\hbar\omega) \propto \frac{k_r(\hbar\omega)}{\Phi_{BB}(\hbar\omega)}. \text{ (M5.1)}$$

Supplementary Equation (M5.1) correlates the emission ($k_r(\hbar\omega)$) to absorptance ($A(\hbar\omega)$). Another way of examining detailed balance is to utilize the absorption coefficient of CT states ($\alpha_{abs}^{CT}(\hbar\omega)$), which is more convenient using our model. Here, we rewrite Supplementary Equation (M4-3) using a Taylor expansion as:

$$\begin{aligned}\alpha_{abs}^{CT}(\hbar\omega) &= -\frac{1}{2d}\ln(1-A) \\ &= -\frac{1}{2d}\left(-A - \frac{A^2}{2} - \frac{A^3}{3} - \frac{A^4}{4} - \dots\right),\end{aligned}\quad (M5.2)$$

In the case of absorptance from CT states, A is much smaller than 1. Therefore, in order to obey detailed balance, the proportionality shown below should apply:

$$\alpha_{abs}^{CT}(\hbar\omega) \approx \frac{A(\hbar\omega)}{2d} \propto A(\hbar\omega) \propto \frac{k_r(\hbar\omega)}{\Phi_{BB}(\hbar\omega)},\quad (M5.3)$$

Therefore, both Supplementary Equation (M5.1) and Supplementary Equation (M5.3) can be used to test the principle of detailed balance. Practically, $A(\hbar\omega)$ is easier to measure, therefore, Supplementary Equation (M5.1) is the most frequently used method in experiments assuming the device is not limited by charge collection, i.e. $A(\hbar\omega) \sim EQE(\hbar\omega)$. However, in the simulations, we find that Supplementary Equation (M5.3) offers a better way to test the principle of detailed balance, as we can expand the fitting range to the full photon energy spectrum (examples can be seen in Supplementary Fig. 2).

Comment 12:

12) It seems that the simulations in Fig. 2 are still based on a single Gaussian disorder model, is this true? If so, it is confusing to start with the “old model” and a clearer way of presentation is required.

Responses:

Thank you for this question.

The reviewer is right, the original Figure 2 showed simulations based on single Gaussian density of states within our model framework. The reason we use single gaussian density of states of CT states is because this has been widely used as the model “Density of States” (DoS) in the field of organic solar cells. To help demonstrate the capability of the model we have now added more simulations to this figure showing more complex density of states functions. In

addition to the single gaussian CT manifold, we also illustrate two cases of static disorder in a device, in which we include two CT manifolds with either varied distribution width or varied energy of lower CT manifold. We have also revised the relevant part in the general model results section (see changes in blue) as well as the new Fig. 2, Fig. 3, Supplementary Table 2, and Supplementary Table 3. See below the changes:

Changes in line 175-239 in the main text (changes are in blue):

“We firstly explore three variations of $g(E_{CT})$, as shown in Fig. 2a,e,i, a single gaussian CT manifold with varied distribution width (Fig. 2a, DoS 1), two gaussian CT manifolds with varied distribution width (Fig. 2e, DoS 2), and two gaussian CT manifolds with varied peak energy of the lower energy CT manifold (Fig. 2i, DoS 3). As a first step, we show that our model obeys the principle of detailed balance, which links the absorption and emission in a solar cell through the ambient black body radiation flux (ϕ_{BB}),⁵³ and which has been shown to be obeyed in most inorganic⁵⁴⁻⁵⁶ and organic^{57,58} semiconductor based photovoltaics. The equations underpinning detailed balance are given by Supplementary Equation (M5.1).⁵³ Shown in Fig. 2b,f,j are the calculated and normalized absolute emission rate (k_r) per unit photon energy ($\hbar\omega$), the absorptance (A), and the ratio k_r/ϕ_{BB} , for three types of $g(E_{CT})$). The perfect overlap between the tail of A and k_r/ϕ_{BB} in the presence of static disorder demonstrates the validity of the detailed balance principle using our model. The parameters used to produce Fig. 2 are presented in Supplementary Table 1-3, and we note here that neither the choice of the values of those parameters nor the shape of $g(E_{CT})$ should violate the principle of detailed balance using our model (see also Supplementary Fig. 2).

We now analyse the effect of different levels of static disorder (through varying $g(E_{CT})$) on the absorption and emission profiles of photovoltaics and finally on the voltage losses (Fig. 2). We carry out a simulation assuming that all the other CT-state parameters are preserved while changing $g(E_{CT})$ using a typical set of CT-state

parameters (given in Supplementary Table 1-3). We note that this assumption is unlikely to hold in real devices, but it is a useful and necessary first approximation to model the effect of static disorder. The effects of increasing static disorder on the absorption and emission profiles of the device are shown in Fig. 2b,f,j. The model shows that the low-energy part of $A(\hbar\omega)$ becomes broader with broadened $g(E_{CT})$, whilst the peak of the emission spectrum shifts to the red. This is reflected in the total (integrated over energy) rate constant of radiative recombination (K_r , Fig. 2c,g,k), which decreases slightly with broadened $g(E_{CT})$, in accordance with the energy gap law, which is embodied in our model. Concomitantly, as the CT states spread out in energy, the proportion of nonradiative transition from lower energy CT state to ground state is increased, resulting in higher values of the energy-integrated non-radiative rate constant K_{nr} as also shown in Fig. 2c,g,k. From the rates of radiative and non-radiative recombination, we can then calculate the voltage loss due to non-radiative recombination, ΔV_{nr} , which is the main voltage loss contribution to the voltage loss in OPV^{39,58}. Clearly, as shown in Fig. 2d, 2h and 2l, increased static disorder (i.e. as $g(E_{CT})$ is broadened) is detrimental to both ΔV_{nr} , which increases and $V_{oc,rad}$, which decreases. $V_{oc,rad}$ is the radiative limit of V_{oc} and represents the maximum open-circuit potential available for a device of given absorption profile and, since it can be defined precisely for any absorption spectrum, is a more useful quantity than E_g ⁵⁹ and E_{CT} ¹³. The decreased $V_{oc,rad}$ with increasing σ_{CT} can be rationalized by the broadened absorption edge, which leads to more radiative losses according to the reciprocity relation⁵³, while increased ΔV_{nr} with σ_{CT} is caused by enlarged K_{nr} and reduced K_r (Supplementary Equation (M2.3) and (M2.4)). This leads to a negative dependence of ΔV_{nr} on $V_{oc,rad}$, as shown in Fig. 3a for the three types of $g(E_{CT})$ considered, where ΔV_{nr} is plotted against $V_{oc,rad}$.^{8,38,58} We also note here that the extra nonradiative voltage losses caused by static disorder that were predicted by Burke et al.⁹ (i.e. $\sigma_{CT}/2k_B T$) using a gaussian distribution of CT states and without considering the energy

gap law are overestimated as compared to our model results (see Supplementary Fig. 3).

Using the single gaussian CT state distribution as an example, we now go on to model the effects of different CT-state parameters that affect CT-state recombination as a function of static disorder, as in our previous work⁸; the effects of varying high- and low-frequency reorganization energies (λ_i and λ_o), the CT-state to ground state oscillator strength (f_{osc}), E_{CT} , and the ratio of CT to local exciton state densities ($N_{CT/LE}$) on ΔV_{nr} and $V_{oc,rad}$ are thus plotted in Fig. 3b, for two different σ_{CT} , i.e. $\sigma_{CT} = 0$ and $\sigma_{CT} = 0.1$ eV (further details are in Supplementary Fig. 4-8). The results show that the effect of each of the parameters (for further explanation refer to Ref. ⁸), is qualitatively preserved regardless of the level of static disorder present in the device, since changing σ_{CT} from 0 to 0.1 eV, as shown in Fig. 3b, shifts the whole plot to higher ΔV_{nr} and lower $V_{oc,rad}$. However, the magnitude of the effect of changing each of the other parameters changes at different σ_{CT} , i.e. increasing σ_{CT} lowers the impact of λ_o and λ_i , but enlarges the impact of f_{osc} , E_{CT} , and $N_{CT/LE}$ (see Supplementary Fig. 4-8 for details). The lower impact of reorganisation energy suggests that if static disorder is dominant, the impact of dynamic disorder is then reduced. Clearly, static disorder can have a major impact on the analysis of voltage loss. We also compare our model with the trend in voltage loss predicted by Benduhn et al.⁷ with experimental data taken from Ref. ⁶⁰ as well as data collected in this work (Fig. 3b). It is evident that the introduction of static disorder influences the relationship between the material parameters and ΔV_{nr} and $V_{oc,rad}$. We further discuss the effects of static disorder on the voltage and efficiency limit in Supplementary Note 2, noting here that increasing σ_{CT} from 0 to 0.1 eV can affect the relationship between V_{oc} and E_g/q (hence PCE vs. E_g/q relation) (see Fig. 3c,d), and reduce both the voltage and efficiency by ~33% according to our model.”

New Figure 2 in the main text:

Fig. 2. General model results on the effect of static disorder. Three types of $g(E_{CT})$ are shown: **a** DoS 1: Single gaussian CT state manifold with varied σ_{CT} ; **e** DoS 2: Two gaussian CT manifolds with varied σ_{CT} (two CT manifolds have the same σ_{CT} and varies simultaneously); **i** DoS 3: Two gaussian CT manifolds with $E_{CT,C1}$ (lower energy CT manifold) changed while $E_{CT,C2}$ was fixed. Normalized rate per photon energy of emission (k_r), absorptance (A) and k_r/ϕ_{BB} for **b** 1 gaussian CT states with varied σ_{CT} ; **f** 2 gaussian CT manifolds with varied σ_{CT} ; **j** 2 gaussian CT manifolds with varied $E_{CT,C1}$. Rate constant of radiative (K_r) and nonradiative (K_{nr}) transition for **c** 1 gaussian CT states with varied σ_{CT} ; **g** 2 gaussian CT manifolds with varied σ_{CT} ; **k** 2 gaussian CT manifolds with varied $E_{CT,C1}$. ΔV_{nr} and $V_{oc,rad}$ for **d** 1 gaussian CT states with varied σ_{CT} ; **h** 2 gaussian CT manifolds with varied σ_{CT} ; **l** 2 gaussian CT manifolds with varied $E_{CT,C1}$. The range of σ_{CT} is chosen based on experimentally observed values ranging up to ~ 0.1 eV^{9,11}. Yellow shaded area indicates the size of V_{oc} determined by $V_{oc,rad} - \Delta V_{nr}$. The detailed parameters used in this figure are listed in Supplementary Table 1-3.

New Figure 3 in the main text:

Fig. 3. Voltage loss analysis and model comparisons, voltage and efficiency limits and single-state analysis. **a** $\Delta V_{nr}(V_{oc,rad})$ plot generated from Fig. 2 with the three different DoS-CT distributions shown in Figure 2, i.e. a single Gaussian CT state manifold (DOS 1), a Two-Gaussian CT state manifold of varying breadth, and a Two-Gaussian CT state manifold of varying peak energies. Arrows indicate the direction of increasing static disorder. **b** Voltage loss model comparisons using $\Delta V_{nr}(V_{oc,rad})$ plot with varied properties of CT state, including

$\lambda_o, \lambda_i, f_{osc}, E_{CT,center}, N_{CT/LE},$ and σ_{CT} . The ranges for these parameters are $\lambda_o = [0.05, 0.2]$ eV, $\lambda_i = [0.05, 0.2]$ eV, $f_{osc} = [10^{-2}, 1]$, $E_{CT,C} = [1.5, 1.65]$ eV, $N_{CT/LE} = [10^{-3}, 1]$, and $\sigma_{CT} = [0, 0.1]$ eV. Solid lines with circles represent the case with $\sigma_{CT} = 0.0$ eV, while dashed lines with circles represent the case with $\sigma_{CT} = 0.1$ eV. The calculations are compared with the model by Benduhn et al.⁷ (dashed grey line) and a large amount of reported data taken from Ref.⁶⁰ (light-grey scatters) as well as the data collected in this work (light-red stars). The arrows in (e) indicate the direction of increasing variable values. The details of the effect of each parameter on absorption, emission, rate constants, and V_{loss} have been presented in Supplementary Fig. 4-8. The parameters used to produce Supplementary Fig. 4-8 are listed in Supplementary Table 4. Note here the model by Benduhn is plotted as function of $V_{oc,rad}$ using an approximated linear relation that $V_{oc,rad} = 0.833E_{CT}/q$ (see Supplementary Fig. 9). **c** Voltage limits and **d** Efficiency limits as a function of optical gap divided by elementary charge (i.e. E_g/q). Detailed parameters and discussion for voltage and efficiency limits can be found in Supplementary Note 2 and Supplementary Table 5. **e** Example of single-state analysis using reduced emission ($\propto k_r/(\hbar\omega)^3$) and absorption ($\propto A/\hbar\omega$) spectrum. **f** Extracted $E_{CT,eff}$ and λ_{eff} as a function of σ_{CT} using 1-gaussian DoS as an example. Simulated emission and absorption spectrum are used as the input for single-state analysis. Details of the comparisons on the calculated emission, absorption, recombination rate, and voltage losses are shown in Supplementary Fig. 11. The parameters used to produce the input spectrum are listed in Supplementary Table 1.

New Supplementary Table 2 in the SI:

Supplementary Table 2. Key parameter values used in Figure 2e-h

Parameter	Value	Units
Difference in the static dipole moment ($ \overline{\Delta\mu} $)	5	Debye
Oscillator strength (f_{osc})	1×10^{-2}	Unitless
High frequency reorganisation energy (λ_i)	0.2	eV
Low frequency reorganisation energy (λ_o)	0.2	eV
Vibrational mode harmonic oscillator energy ($\hbar\Omega$)	0.15	eV

refractive index (n)	1.5	Unitless
The ratio of CT state density ($R_{CT/LE}$)	1	Unitless
Energy of local excitonic state (E_{LE})	1.8	eV
Energy of the centre of CT states ($E_{CT,C1}$)	1.2	eV
Energy of the centre of CT states ($E_{CT,C2}$)	1.5	eV
Gaussian width for both CT manifold ($\sigma_{CT,C1}$ and $\sigma_{CT,C2}$)	0.0-0.1	eV

Note: light yellow shaded area indicates the variable(s). For simplicity, we assume CT_1 and CT_2 manifold share the same set of parameters, except for the peak energies.

New Supplementary Table 3 in the SI:

Supplementary Table 3. Key parameter values used in Figure 2i-l

Parameter	Value	Units
Difference in the static dipole moment ($ \overline{\Delta\mu} $)	5	Debye
Oscillator strength (f_{osc})	1×10^{-2}	Unitless
High frequency reorganisation energy (λ_i)	0.2	eV
Low frequency reorganisation energy (λ_o)	0.2	eV
Vibrational mode harmonic oscillator energy ($\hbar\Omega$)	0.15	eV
refractive index (n)	1.5	Unitless
The ratio of CT state density ($R_{CT/LE}$)	1	Unitless
Energy of local excitonic state (E_{LE})	1.8	eV
Energy of the centre of CT states ($E_{CT,C1}$)	1-1.4	eV
Energy of the centre of CT states ($E_{CT,C2}$)	1.5	eV
Gaussian width for both CT manifold ($\sigma_{CT,C1}$ and $\sigma_{CT,C2}$)	0.05	eV

Note: light yellow shaded area indicates the variable(s). For simplicity, we assume CT_1 and CT_2 manifold share the same set of parameters, except for the peak energies.

Comment 13:

13) What do the authors mean by saying they check whether their “results obey the principle of detailed balance”? From the text, I understood that they enforce this in the model. Clarification and explanation is required.

Responses:

Thank you for this question. The reviewer is correct that the principle of detailed balance is enforced in the model and should not have to be demonstrated. We chose to demonstrate that the principle of detailed balance is obeyed in our model using Figure 2 partly in order to demonstrate to other groups in visual way, how a model could be tested. We note that demonstrating this agreement takes up no additional journal space. As noted above we have now added less trivial cases containing two CT state manifolds to both Figure 2 and Figure 3a, reproduced above, which may be of more general interest. We discuss in detail how we can test the detailed balance principle experimentally in Supplementary Method 5 in the SI.

Comment 14:

14) I don't see why the validity of their model depends on the principle of detailed balance? The assumptions underlying the model should be stated more clearly.

Responses:

Thank you for raising this concern. For our purpose, which is to evaluate the voltage losses in solar cells relative to the radiative limit, it is important that any model used should obey detailed balance. See also (see: Nature Materials 13, 63–68(2014) and Phys. Rev. Applied 4, 014020 (2015)) for discussion of the physical relevance of detailed balance for the operation of various types of photovoltaic devices, including organic solar cells. Regarding the assumptions in the model, we have extended the text accordingly as explained in the responses to reviewers' points 2-5.

Comment 15:

15) It seems that Fig. 2a and 2b show the same results. Why are they needed both?

Responses:

Thank you for asking this question. In fact, they are not the same results. As we have discussed above on the methods of testing the principle of detailed balance, we can use either use the absorption coefficient of CT states ($\alpha_{abs}^{CT}(\hbar\omega)$) or the tail of absorptance ($A(\hbar\omega)$) to do so following Supplementary Equation (M5) in the SI. As we also explained in the responses to **Comment 11** and in Supplementary Method 5 in the SI, practically $A(\hbar\omega)$ is more frequently used in the literature since one can measure it as the external quantum efficiency. However, in the model, we find $\alpha_{abs}^{CT}(\hbar\omega)$ is more straightforward since it can be calculated easily and offers a wider range of spectrum as a function of photon energy instead of just the tail of $A(\hbar\omega)$. Therefore, in the previous version both were shown in Figure 2. The new text which has been added to Supplementary Method 5 in the SI is intended to explain why both presentations are useful.

However, we agree that showing both of might not be necessary in the main text, therefore in the revised Fig. 2, we only show $A(\hbar\omega) \sim k_r / \phi_{BB}$.

Comment 16:

16) Same data is labeled kr in one subfigure and kr/Φ_{BB} in the other.

Responses:

Thank you. This has been corrected in the revised Fig. 2.

Comment 17:

17) Figure 2d is not mentioned in the text.

Responses:

Thank you for pointing this out, the previous Figure 2d is now mentioned along with Figure parts 2d, 2h and 2l in the revised manuscript (line 206 in page 9):

“Clearly, as shown in Fig. 2d, 2h and 2l, increased static disorder (i.e. as $g(E_{CT})$ is broadened) is detrimental to both ΔV_{nr} , which increases and $V_{oc,rad}$, which decreases.”

Comment 18:

18) As a general remark, I find the discussion of the results not suitable for the general and the expert reader, because it is not explained in an accessible way. This makes it extremely hard for the reader to grasp the significance of the work. The discussion of the results is very superficial. Sometimes the formulas that are used to obtain the results are not mentioned. Sometimes only one sentence is written and the authors proceed to the next part.

Responses:

We thank the reviewer for his or her critical reading of the paper and apologize for the confusion caused. We have revised the text of the results section in the main text giving more attention to discussion of results and the formulae used to obtain them. The main improvements in the manuscript are shown in the responses to **Comment 2-5, 7-15, 19-25, 29, and 35**.

Comment 19:

19) What do we learn from Figure 2e? Why the comparison to Ref. 7?

Responses:

Thank you for raising this question. Figure 2e (i.e. Fig. 3b in the revised version) is the summarized simulation results based on Supplementary Fig. 4-8, where the effect of different CT state properties, for example, oscillator strength, were simulated with or without static disorder using the proposed model. These simulations were partly motivated by our previous model results without any static disorder that has been published in 2018 (see: PHYS. REV. X 8, 031055 (2018)) to have an idea of whether our previous modeling results are still valid, and the answer is yes, they are qualitatively valid. However, the model results predict that the introduction of static disorder could reduce the impact of reorganization energies, yet enhance the impact of other parameters, for example, oscillator strength.

The comparison with Ref. 7, which is the paper by Benduhn et al., is first to show that when static disorder is involved, the model in Ref. 7 is oversimplified, then to show that static disorder offers another way to explain those scatter points in Figure 2e (i.e. Fig. 3b in the revised version).

We believe that the main message as we just discussed above has been given in the original manuscript, and from line 219-239 on page 10 in the main text in the revised version (see also below):

“Using the single gaussian CT state distribution as an example, we now go on to model the effects of different CT-state parameters that affect CT-state recombination as a function of static disorder, as in our previous work⁸; the effects of varying high- and low-frequency reorganization energies (λ_i and λ_o), the CT-state to ground state oscillator strength (f_{osc}), E_{CT} , and the ratio of CT to local exciton state densities ($N_{CT/LE}$) on ΔV_{nr} and $V_{oc,rad}$ are thus plotted in Fig. 3b, for two different σ_{CT} , i.e. $\sigma_{CT} = 0$ and $\sigma_{CT} = 0.1$ eV (further details are in Supplementary Fig. 4-8). The results show that the effect of each of the parameters (for further explanation refer to Ref. ⁸), is qualitatively preserved regardless of the level of static disorder present in the device, since changing σ_{CT} from 0 to 0.1 eV, as shown in Fig. 3b, shifts the whole plot to higher ΔV_{nr} and lower $V_{oc,rad}$. However, the magnitude of the effect of changing each of the other parameters changes at different σ_{CT} , i.e. increasing σ_{CT} lowers the impact of λ_o and λ_i , but enlarges the impact of f_{osc} , E_{CT} , and $N_{CT/LE}$ (see Supplementary Fig. 4-8 for details). The lower impact of reorganisation energy suggests that if static disorder is dominant, the impact of dynamic disorder is then reduced. Clearly, static disorder can have a major impact on the analysis of voltage loss. We also compare our model with the trend in voltage loss predicted by Benduhn et al.⁷ with experimental data taken from Ref. ⁶⁰ as well as data collected in this work (Fig. 3b). It is evident that the introduction of static disorder influences the relationship between the material parameters and ΔV_{nr} and $V_{oc,rad}$. We further discuss the effects of static disorder on the voltage and efficiency limit in Supplementary Note 2, noting here that increasing σ_{CT} from 0 to 0.1 eV can affect the relationship between V_{oc} and E_g/q (hence PCE vs. E_g/q relation) (see Fig. 3c,d), and reduce both the voltage and efficiency by ~33% according to our model.”

Comment 20 and 21:

20) Figure 4 shows EQE spectra, whose edges are fitted to extract Urbach energies. How can the authors be sure this part is from CT states? Nothing has been mentioned to support this assumption. I am wondering about the trend with increasing wt% in both cases Fig. 4a and Fig. 4d. Why does the CT signal decrease with higher NFA content?

21) The authors just say that the result was interesting and state the numbers for the Urbach energies for the blends. But what does the EQE data contribute to the MS and the understanding of these systems?

Responses:

We appreciate the two questions regarding the analysis of EQE data.

A simple method to determine whether the absorption tail is from CT state or not is to compare the absorption tail of blends to the absorption tail of the pristine material of the lower optical gap, which is the NFA in each case. Here, we have added the absorption from both pristine materials in Figure 4a and 4d, where the clear difference between blend's absorption tail and pristine material's absorption tail can be seen. This difference means the absorption tails of the blends in Figure 4a and 4d must arise from CT states. We have also added some text in the main text and Figure 4 caption to mention this (see also below in blue):

“Focusing first on the EQE edges, we can identify clear CT state absorption in all the blends, as it can be clearly differentiated from the singlet absorption of the pristine donor and acceptor, as displayed in Fig. 4a and 4d.”

And the figure caption of Figure 4:

Fig. 4. Experimental EQE, low injection EL and V_{loss} . Normalized EQE of **a** O-IDTBR and **(d)** O-IDFBR devices; Normalized EL of **b** O-IDTBR and **e** O-IDFBR devices; and $\Delta V_{nr}(V_{oc,rad})$ plot for **c** O-IDTBR and **f** O-IDFBR devices. wt% of 20%, 40%, and 70% are chosen as representative devices. Urbach energy E_U and the range of fittings are displayed in **a** and **d**. EQE and Photoluminescence (PL) spectra of pristine materials are indicated as black or grey lines. The emission from LE state in the O-IDTBR and O-IDFBR blends are indicated by the PL from pristine O-IDTBR and P3HT (the lower band gap component), respectively. Note here that the injection current for EL experiments is chosen to be low enough to ensure the devices are close to QTE and to be the same for different NFA wt%. EQE due to CT states is evident in the tail of the blend spectra and can be clearly distinguished from the EQE of the pristine material that is due the absorption of LE states. This means the absorption tail can be assigned to the absorption of CT states in both O-IDTBR and O-IDFBR cases. The error bars in **c** and **f** indicate the standard derivation.

Regarding the reviewer's query as to why the CT signal decreases with higher NFA content, we first remark that the reasons for the changed absorption tail shape for O-IDTBR and O-IDFBR devices when more NFA is added are different, as we have discussed in detail in the original manuscript. Here, we briefly explain the difference. For the case of O-IDTBR, no

evidence has been found that density distribution of CT states changes upon varying the NFA weight fraction. However, with more O-IDTBR included, we suggest that the crystal size of O-IDTBR grows, which has been shown to reduce the electronic coupling, hence oscillator strength (see examples: ACS Nano 2016, 10, 8, 7619–7626, and PHYS. REV. X 8, 031055 (2018)), therefore would explain the reduction of CT absorption intensity. Differently, in the case of O-IDFBR, we find that the contribution from the lower energy, CT₁, manifold is significantly reduced when more O-IDFBR is added in the blend. This also leads to a change in CT absorption tail slope with increasing NFA content but for a different reason than in the case of O-IDTBR.

In addition, we here explain why we present EQE here. Basically, EQE only served as the first evidence that the density distribution of CT states might change as a function of NFA weight fraction in the case of O-IDFBR. We cannot draw solid conclusion based on just EQE, therefore, later (from Line 300 to 356 in the main text), we use EL data to explain in detail how the changes in EQE tail are induced with the aid of a recent detailed study of the phase behavior of the blends we studied in this manuscript (please see: Chem. Mater. 32, 8294–8305 (2020)). With the help of both EQE and EL, we can estimate the energetic distribution of CT states. Then, eventually, we use our model to explain the experimental results. A summary of the key findings in the Ref., Chem. Mater. 32, 8294–8305 (2020), that relate to the present manuscript is provided in the manuscript at the beginning of the experimental section from line 265-280.

The text copied from line 264-280 in the main text:

“Relating phase behaviour to static disorder. For the purpose of testing the utility of our model, we now proceed to use it to explain the experimental results of two OPV blends, namely P3HT: ((5Z,5'Z)-5,5'-(((4,4,9,9-tetraoctyl-4,9-dihydro-s-indaceno[1,2-b:5,6-b']dithiophene-2,7-diyl) bis(benzo[c][1,2,5]thiadiazole-7,4-diyl) bis (methanylylidene)) bis (3-ethyl-2-thioxothiazolidin-4-one)) (O-IDTBR)⁶¹ and P3HT: (5Z,5'Z)-5,5'-((7,7'-(6,6,12,12-tetraoctyl-6,12-dihydroindeno[1,2-b]fluorene-2,8-diyl)bis(benzo[c][1,2,5]thiadiazole-7,4-diyl))bis(methanylylidene))bis(3-ethyl-2-thioxothiazolidin-4-one) (O-IDFBR)⁶². The chemical structures of the molecules are shown in Supplementary Fig. 14. The two materials are particularly interesting because they are chemically very similar but due to the difference in

the molecular planarity, the planar O-IDTBR shows an increased propensity to crystallize in blends than the twisted O-IDFBR⁶³. Using detailed optical probes, we have previously shown that in blends with P3HT, with O-IDTBR the crystallinities of both P3HT and O-IDTBR are largely preserved over a wide range of blend ratios⁶³, whereas for the less crystalline O-IDFBR blends P3HT crystallinity is easily disrupted by introducing more O-IDFBR⁶³, leading to more amorphous interfaces at higher O-IDFBR content blends. The contrasting behaviour of P3HT:O-IDFBR and P3HT:O-IDTBR as a function of composition^{29,32,33} allows us to fabricate two sets of devices with different underlying $g(E_{CT})$.”

We however apologize for the confusion caused by insufficient explanation of the EQE data. Here we have added some discussion in line 294 to 297 on page 12 of in the main text:

“This suggests that the DoS-CT distribution of the lower energy CT manifold likely remains unchanged for the O-IDTBR devices but changes for the O-IDFBR devices when NFA wt% is varied, a conclusion which is supported by the EL spectra (Fig. 4 b and e), discussed below.”

Comment 22:

22) *Figure 4c and 4f are not discussed beyond simply stating what the reader can see anyhow in the figure.*

Responses:

Following this comment, we have added more discussion in the relevant part (see the changes in blue):

Line 359 to 375 in page 14 in the main text:

“**Voltage loss analysis.** Using EQE and EL under low injection, we can then quantify the voltage loss by calculating $V_{oc,rad}$ and ΔV_{nr} using the method presented in Supplementary Method 2 (or Ref. ⁵⁸). We note that our method to experimentally quantify ΔV_{nr} (see

Supplementary Method 2) remains valid given the principle of detailed balance is fulfilled⁵³ with or without static disorder. Different trends as a function of composition can be seen in Fig. 4c and 4f. In both cases, we see increased $V_{oc,rad}$ with increasing NFA content, which can be explained by the reduced absorption by the CT state leading to less radiative recombination (Fig. 4a,d).⁵⁸ The resulting ΔV_{nr} increases with $V_{oc,rad}$ upon increasing O-IDTBR wt%. (Fig. 4c), whereas ΔV_{nr} reaches a minimum at 40% (wt%) NFA for the O-IDFBR devices (Fig. 4f). The increase of ΔV_{nr} as a function of O-IDTBR wt% can be rationalized by the reduced emission intensity, as observed in the EL spectrum (Fig. 4b). According to Supplementary Equation (M2.3), less emission would lead to lower EQE_{EL} and therefore higher ΔV_{nr} .⁵³ For O-IDFBR devices, the reduction of ΔV_{nr} can be explained by the reduced contribution of nonradiative recombination from lower energy CT states following the energy gap law⁷, as can be seen from the EL in Fig. 4e. We finally note that the large ΔV_{nr} in 70% O-IDFBR device suggests a different mechanism may be involved for this particular blend, which cannot be explained by simply considering the reduced contribution from lower energy CT states. The complete device and voltage loss data are presented in Supplementary Fig. 17 and Fig. 18.”

Comment 23:

23) *A strong assumption in the paper is about the origin of the emission peaks at low energies (Fig. 5). This is the basis for parts of the discussions. The authors state “We suggest that the two peaks are induced by interfaces with semi-crystalline polymer (lower energy CT1) and interfaces with amorphous polymer (higher energy CT2)”. This seems to remain a difficult point to verify, despite the EL data at hand. Is there additional evidence that can be used to consolidate this assumption?*

Responses:

Thank you for this question.

Actually, the idea of using the blends of O-IDTBR or O-IDFBR with P3HT as our testing devices for our model came from our recent study on the phase behavior of P3HT:O-IDTBR and P3HT:O-IDFBR blends as a function of composition (see: Chem. Mater. 32, 8294–8305 (2020)). In that paper, we carried out various experimental characterization methods (including Differential scanning calorimetry, Raman spectroscopy, ellipsometry and photoluminescence) to understand the phase behavior upon changing NFA weight fraction, where we can differentiate the semi-crystalline and amorphous interface. This aids to the proposal of different CT states corresponding to different interfaces.

To make it clearer, we have added a few words in line 321 in page 13 in the main text:

“We suggest that the two peaks are induced by interfaces with semi-crystalline polymer (lower energy CT₁) and interfaces with amorphous polymer (higher energy CT₂) in accordance with our previous findings (Ref⁶³) on the phase behaviour of the two systems.”

Comment 24:

24) Why does CT2 disappear in Fig. 5b with decreasing P3HT wt%?

Responses:

Thank you for asking this question.

Basically, when we put in more O-IDTBR, due to the strong tendency of O-IDTBR to crystallize, clusters of O-IDTBR grow (please see: Chem. Mater. 32, 8294–8305 (2020)). As a result of the increasing acceptor domain size, a greater proportion of excitons cannot diffuse to the donor:acceptor interfaces, therefore we see increased emission from localized excitonic states relative to that from CT states. The strong emission then overlaps with CT₂ emission. Due to the low oscillator strength of CT states in general and given also the small energy difference between CT₂ and LE state, emission from CT₂ is not discernable in high NFA weight fraction devices.

We have also revised the text in line 337 in the main text to make it clearer:

“Upon introducing more O-IDTBR in the blends, the reduced intensity of CT₁ can be assigned to the aggregation and crystallization of O-

IDTBR and the associated loss of interface area and electronic coupling^{8,29}; emission from the O-IDTBR exciton dominates the spectrum in the 40% and 70% devices and therefore we cannot clearly observe CT₂ in the 40% device or either CT manifold in the 70% device.”

Comment 25:

25) *The authors continue with a two-state CT model (Fig. 6) in which additional parameters enter to fit the experimental data. The connection of the model parameters such as modification in f_{osc} or relative weights of CT1 and CT2 should be better motivated and explained. This is not very transparent. How are F_{C1} and F_{C2} defined?*

Responses:

Thank you for the concern raised here.

Basically, the variables in the modelling section were chosen based on the experimental observed changes in EQE tail, and CT state emission. As we have discussed in the manuscript, we see a clear intensity drop of both EQE-tail and EL intensity from CT states when more O-IDTBR was added in the blend. The addition of O-IDTBR gives rise to the increased crystalline domain (cluster) of O-IDTBR (see Chem. Mater. 32, 8294–8305 (2020)), which has been shown to reduce the electronic coupling elements (see ACS Nano 2016, 10, 8, 7619–7626). Therefore, we rationalized the changes by the changes on the oscillator strength of CT states for the case of O-IDTBR in our model. In the calculations we varied f_{osc} of CT state in the range of $[10^{-3}, 10^{-1}]$. The range of f_{osc} was chosen based on the change of the relative CT state emission intensity from Figure 4b, where roughly two orders of magnitude change was seen.

In the case of O-IDFBR, we observed a clear intensity drop of lower energy CT manifold in the emission spectrum (Fig. 4e) as we add more O-IDFBR in the blend. As discussed in the manuscript in detail, the observed lower energy CT states was believed to come from the semi-crystalline P3HT: amorphous O-IDFBR interfaces, whose density fraction decreases with O-IDFBR w.t. (see also Chem. Mater. 32, 8294–8305 (2020) for the phase behavior study). And, F_{C1} and F_{C2} are the density fraction of semi-crystalline P3HT: amorphous O-IDFBR interfaces and amorphous P3HT: amorphous O-IDFBR interfaces, and in the calculations, we assume the density fraction of CT₁ (i.e. F_{C1}) changes with F_{C2} unchanged. And we vary F_{C1} in the range

of [0.01, 0.7]. The range of F_{C1} was chosen based on the estimated interfacial density fraction changed based on Ref. Chem. Mater. 32, 8294–8305 (2020), where ~70% of crystalline interface was observed in the case of 20% O-IDFBR devices. This value was significantly reduced when more O-IDFBR was added in the blend. We here chose to have roughly two orders of magnitude changes to see the effect clearly.

Therefore, both variables (f_{osc} and F_{C1}) have been justified in the manuscript. We have also revised Supplementary Table 6 and the notes to make it clearer (see also below):

Supplementary Table 6. Key parameter values to model experimental devices in Fig. 6

Parameter	O-IDTBR		O-IDFBR		Units
	CT ₁	CT ₂	CT ₁	CT ₂	
Difference in the static dipole moment ($ \overline{\Delta\mu} $)		5			Debye
High frequency reorganisation energy (λ_i)		0.2			eV
Low frequency reorganisation energy (λ_o)		0.2			eV
Vibrational mode harmonic oscillator energy ($\hbar\Omega$)		0.15			eV
refractive index (n)		1.5			Unitless
The ratio of CT state density ($R_{CT/LE}$)		0.01			Unitless
Energy of local excitonic state (E_{LE})		1.7		1.9	eV
Oscillator strength (f_{osc})		[10^{-3} , 10^{-1}]	10^{-3}	10^{-1}	Unitless
The density fraction of CT state (F_C)	0.7	0.3	[0.01, 0.7]	0.3	Unitless
Energy of the centre of CT states ($E_{CT,C}$)	1.2	1.5	1.3	1.5	eV
CT DOS width (σ_{CT})	0.02	0.04	0.02	0.04	eV

Note: E_{LE} is estimated using Supplementary Fig. 20, and for simplicity in the modelling we assume it's unchanged while changing the composition as it's not the determining factor. The range of f_{osc} was chosen based on the change of the relative CT state emission intensity from Figure 4b, where roughly two orders of magnitude change was seen. The range of F_{C1} was chosen based on the estimated interfacial density fraction changed based on Ref. Chem. Mater. 32, 8294–8305 (2020), where ~70% of crystalline interface was observed in the case of 20% O-IDFBR devices. This value was significantly reduced when more O-IDFBR was added in the blend. We here chose to have roughly two orders of magnitude changes to see the effect clearly.

Comment 26:

26) Fig. 6a and 6b are not clear about the parameter on the y-axis, is this the NFA content (which range)? Additional assumptions should be made clearer in a different way.

Responses:

We agree this needs clarifying.

The y-axis in the case of O-IDTBR (now in Supplementary Fig. 21a) is oscillator strength of CT state, i.e. f_{osc} . For O-IDFBR (now in Figure 6a), the y-axis is the density fraction of CT₁, i.e. F_{CT1} . They have been labelled. And the direction of change of the parameters is to present the increase of NFA content in the blends, i.e. from 20% to 70%. To increase the readability, we have revised the figures and added some text in the figure caption, see below in blue:

(note that the simulation of the O-IDTBR device have been moved to the SI following the later comment: **Comment 34**)

Figure 6 in the main text:

Fig. 6. Estimated distribution of density of CT state $g(E_{CT})$, and simulated EQE, EL and V_{loss} for the O-IDFBR devices. Illustration of the distribution of $g(E_{CT})$ in (a) the O-IDFBR devices used in the model. In (a) the trend in $g(E_{CT})$ with increasing O-IDFBR fraction from 20 to 70wt% is modelled via a decrease in the density fraction, F_{CT1} , of the lower energy CT state from 0.7 to 0.01. $g(E_{CT})$ is modelled using of two CT manifolds with a gaussian width of 0.02 eV (semi-crystalline CT₁) and 0.04 eV (amorphous CT₂), respectively. The total $g(E_{CT})$ is a superposition of CT₁ and CT₂. Parts b-d show simulated and normalized injection dependent emission spectra of O-IDFBR devices with different NFA wt%. Blue arrows are indications of increasing chemical potential in the range of 1-2 eV. Parts e-g show absorbance (A), rate of emission per photon energy (k_r), and $\Delta V_{nr}(V_{oc,rad})$ plot. The detailed parameters used to model the O-IDFBR devices are listed in Supplementary Table 6.

Supplementary Fig. 21 in the SI:

Supplementary Fig. 21. Estimated distribution of density of CT state $g(E_{CT})$, and simulated EQE, EL and V_{loss} for the O-IDTBR devices. Illustration of the distribution of $g(E_{CT})$ in (a) the O-IDTBR devices used in the model. In (a), the trend in $g(E_{CT})$ with increasing O-IDTBR fraction from 20 to 70wt% is modelled via a decrease on the oscillator strength f_{osc} of both CT states from 10^{-1} to 10^{-3} . $g(E_{CT})$ is modelled using of two CT manifolds with a gaussian width of 0.02 eV (semi-crystalline CT₁) and 0.04 eV (amorphous CT₂), respectively. The total $g(E_{CT})$ is a superposition of CT₁ and CT₂. Parts b-d show simulated and normalized injection dependent emission spectra with different NFA wt%. Blue arrows are indications of increasing chemical potential in the range of 1-2 eV. Parts e-g show absorbance (A), rate of emission per photon energy (k_r), and $\Delta V_{nr}(V_{oc,rad})$ plot. We note here that the emission from LE state hasn't been simulated as for large D-A HOMO-HOMO offset system (i.e. ~ 0.5 eV for P3HT:O-IDTBR) the contribution from LE state to total recombination is negligible.³⁰ The reason that we can see clear LE state emission from O-IDTBR excitons is because of the large amount of O-IDTBR cluster in the case of high O-IDTBR weight fraction in the blend (i.e. 40% and 70%), this leads to the emission from "isolated" O-IDTBR clusters, which however doesn't contribute to the CT states at all, as they decay directly without charge transfer processes. The detailed parameters used to model the O-IDTBR devices are listed in Supplementary Table 6.

Comment 27:

27) In the models for Fig. 6, is there any LE state? Nothing has been mentioned or explained in the theory section.

Responses:

Thank you for this question.

In the original manuscript, we did include the LE state to better reproduce the EL emission spectra in the case of O-IDTBR devices but did not discuss it since its impact on the result was negligible. After considering the reviewer's comment, we have decided to omit the LE states from the spectral simulations of O-IDTBR devices in Figure 6 (now in Supplementary Figure S21 in the revised version), to simplify the narrative.

The reason to neglect the LE states is that since P3HT:O-IDTBR blends present a large D-A HOMO-HOMO offset (~ 0.5 eV), the LE state should have little influence on the recombination via CT states (see: Nature Energy volume 5, pages 711–719 (2020)), therefore, it's not essential to simulate LE states. The reason that we can see clear LE state emission from O-IDTBR excitons is because of the large amount of O-IDTBR cluster in the case of high O-IDTBR weight fraction in the blend (i.e. 40% and 70%), this leads to the emission from "isolated" O-IDTBR clusters, which however doesn't contribute to the CT states at all, as they decay directly without charge transfer process.

With this being said, we have decided to remove the simulations about the LE states in Figure 6 (now in Supplementary Figure S21 in the revised version), which also helps to reduce the amount of data in the manuscript. We have also added some text in the figure caption of Supplementary Fig. 21 (see also below):

“We note here that the emission from LE state hasn't been simulated as for large D-A HOMO-HOMO offset system (i.e. ~ 0.5 eV for P3HT:O-IDTBR) the contribution from LE state to total recombination is negligible.³⁰ The reason that we can see clear LE state emission from O-IDTBR excitons is because of the large amount of O-IDTBR cluster in the case of high O-IDTBR weight fraction in the blend (i.e. 40% and 70%), this leads to the emission from "isolated" O-IDTBR clusters, which however doesn't contribute to the CT states at all, as they decay directly without charge transfer processes.”

Comment 28:

28) Please clarify how the specified voltage enters.

Responses:

Thank you for asking this. This voltage in the original draft can be understood as the chemical potential energy relative to equilibrium, μ , divided by the elementary charge, i.e. $V = \mu/q$. This quantity was used to represent different levels of applied bias in the injection dependent EL simulations.

However, to be clearer, we have changed the voltage to chemical potential energy to be more accessible, and added a few words in the main text (line 389):

“We first model the injection dependence of the EL spectrum (from Fig. 5) on $g(E_{CT})$ at different compositions (i.e. O-IDFBR wt%) by varying the chemical potential energy (μ) in Equation (5b), noting that the Boltzmann occupation function is now replaced by Fermi-Dirac statistics (i.e. $\{\exp[(m\hbar\Omega_t + E_{CT} - \mu)/k_B T] + 1\}^{-1}$).”

And in the figure caption of Figure 6 we added:

“Blue arrows are indications of increasing chemical potential energy in the range of 1-2 eV.”

Comment 29:

29) In Fig. 6, the differences in the results between Fig. 6a and Fig. 6b and the connection to the differences in the modelling of both NFA compounds is not discussed. Only the curves are described.

Responses:

Thank you for raising this concern. In the revised version, following also **Comment 34**, we have moved the simulation part of O-IDTBR devices into the SI (Supplementary Note 3), the relevant part in the main text and SI have also been revised to provide more details:

Line 376-414 in page 15 in the main text:

“**Reproducing the experimental trends using the static disorder model.** Our model for the variation in the CT state distribution for

P3HT:O-IDTBR and P3HT:O-IDFBR devices as a function of composition is illustrated in Supplementary Fig. 19 and summarized in Supplementary Fig. 21a for the O-IDTBR and Fig. 6a the O-IDFBR devices. The emission, absorption, and voltage loss behaviour of O-IDTBR devices can be well explained by changes in f_{osc} as a function of composition, without any change in $g(E_{CT})$ (see Supplementary Note 3 for simulation results). Such a loss in CT state brightness can be expected from the enlargement of the pure donor and acceptor domains.^{8,29} However, in the case of O-IDFBR, changes in $g(E_{CT})$ upon introducing more O-IDFBR have to be considered in order to rationalize experimental observations as shown below. Such changes or lack of changes in $g(E_{CT})$ for P3HT:O-IDFBR and P3HT:O-IDTBR, respectively, are consistent with the previously identified phase behaviours of these blends⁶³. We model the $g(E_{CT})$ for this blend as two peaks, the lower of which reduces in height, i.e. reducing density fraction of CT₁ (notated as F_{C1} in Fig. 6), as O-IDFBR content is increased.

We first model the injection dependence of the EL spectrum (from Fig. 5) on $g(E_{CT})$ at different compositions (i.e. O-IDFBR wt%) by varying the chemical potential energy (μ) in Equation (5b), noting that the Boltzmann occupation function is now replaced by Fermi-Dirac statistics (i.e. $\{\exp[(m\hbar\Omega_t + E_{CT} - \mu)/k_B T] + 1\}^{-1}$). As shown in Fig. 6 b-d, there is a transition in intensity between two peaks with increased μ in all blend compositions; however as more O-IDFBR is added (40% w.t. and 70% w.t.), the differences between the peaks begin to be washed out as density fraction of CT₁ decreases. For the EQE and low injection EL (Fig. 6 e-f), we can see that the tail of absorptance gets sharper and emission from CT₁ gets weaker as O-IDFBR wt% increases (Fig. 6 e-f), leading to a reduction in ΔV_{nr} . The changes of modelled absorption and emission result from a reducing density fraction of lower energy CT states (i.e. CT₁) in Fig. 6a, and the reduction of ΔV_{nr} results from the reduced non-radiative recombination from CT₁ according to the energy gap law.^{7,44}

These modelling results in Figure 6 and Supplementary Fig. 19 reproduce the key features of the experiments in Fig. 4 and Fig. 5, which cannot be reproduced by a single-CT state model for either material system. The resulting $\Delta V_{nr}(V_{oc,rad})$ plots, Supplementary Fig. 21g can explain the O-IDTBR dependence of voltage losses in that blend while Fig. 6g can explain the change from 20% (wt%) to 40% (wt%) of O-IDFBR, although that it fails to rationalize the change from the 40% (wt%) to the 70% (wt%) O-IDFBR device. We note here that at high O-IDFBR content we enter a regime where hole collection is disrupted⁶³ and that could lead to additional recombination losses due to enhanced non-geminate recombination as well as to an increased rate of CT state decay. However, including this effect is beyond the scope of our model.

We have also explored the impact of changing the energy of the lowest CT state manifold (CT₁) as presented in Supplementary Fig. 22. We note that, in principle, the experimental results could also be fit by an increase in the energy of CT₁ with NFA content. The effect of non-equilibrium site distribution on the validity of our analysis is also commented upon in Supplementary Note 4.”

Supplementary Note 3 in the SI:

“For the O-IDTBR devices, we propose that the position and relative density of both CT₁ and CT₂ remains constant based on the analysis in the main text. However, we suggest that f_{osc} of both CT states reduces as O-IDTBR crystal size grows upon increasing O-IDTBR wt%, based on the reduced emission intensity of CT states in Fig. 4b and previous observation^{6,29}. The simulation results with changed f_{osc} are shown Supplementary Fig. 21.

Starting with injection dependent EL simulations, for all compositions, EL from CT states shows a clear two-peak transition with increased injection, which cannot be reproduced by single state model. We note here we only take account of the emission from CT

states in the simulations, therefore the LE peak in experimental O-IDTBR device with 40% or 70% wt% is not reproduced.

For EQE, low injection EL, and voltage loss simulations, as shown in Fig. 6 e-g, the intensity of the EQE tail and EL emission from CT states reduce, and ΔV_{nr} increases as O-IDTBR wt% increases. This can be rationalized by the reciprocity relation, which relates emission to absorption, and the effect of high CT state emission in reducing ΔV_{nr} .

Key features of experimental observations in Fig. 4 and Fig. 5 for the O-IDTBR devices have been reproduced considering the change of f_{osc} only.”

Comment 30:

30) I find that the “agreement with experiment in Fig. 5” cannot be concluded in this way. From comparing the figures, the differences are obvious.

Responses:

We have revised this sentence in line 401 in page 16 to:

“These modelling results from Figure 6 and Supplementary Fig. 19 thus reproduce the key features of the experiments in Fig. 4 and Fig. 5, which cannot be reproduced by a single-CT state model for either material system.”

Comment 31:

31) The last paragraph on page 14 basically claims qualitative agreement without saying what is meant by this. This is a low level of presentation and needs to be improved. Possibly this is due to the large number of results included in the MS and limited space. Therefore a more specialised journal is more suitable.

Responses:

We appreciate the reviewer’s carefulness and criticism. Therefore, the discussion (from line 376 to line 414) has been revised to explain what we mean by the agreement between model

and experimental data. The changes to this part also address the concern raised in **Comment 29** above, and the revised text is given in our answer to that point.

We have also reduced the amount of results and discussions in the main text by moving the simulations of O-IDTBR to Supplementary Note 3 in the SI, as the O-IDFBR is a better example to test our static disorder model (see also the response to **Comment 29 and 34**).

Comment 32:

32) Coming to the summary, the authors claim “This work presents a model that can be used to quantify and understand the impact of disorder in charge-transfer state energy (static disorder) on voltage losses in organic photovoltaic devices.” Given that the discussion of many aspects of their work is unfortunately very superficial and that in particular the discussion of voltage losses is basically absent from the MS, this claim cannot be sustained.

Responses:

We thank the reviewer for his or her comment. In fact, Section “**General model results**” has a significant amount of discussion on the effects of static disorder on voltage loss analysis. We also hope that the additional discussion that we have added in response to **Comments 22 and 29** provides a more substantial demonstration of how the model can be used to evaluate the impact of static disorder on voltage losses in experimental systems.

Comment 33 and 34:

33) As mentioned in my initial statements, in my opinion this MS suffers from a lack of a clear focus. The statements in the abstract/conclusions are only partly aligned with the text in the MS. It seems that the choice of experimental studies is not so ideal since the P3HT:NFA blends exhibit complicated (and only partly known) phase behavior and quite specific CT state features. At the same time, both systems are also rather similar to each other. In my opinion more space is needed to properly discuss both systems.

34) The relevance of the generality of the results is obscured by the specific properties of the selected systems. It is unclear why both are needed in order to arrive at most of the conclusions presented, such as “We demonstrated that static disorder tends to increase voltage losses in OPV devices, as previously reported, and that it is indeed important in certain well-studied polymer:molecular acceptor material systems, in agreement with Ref. 9,11,17.”

Responses:

We appreciate the reviewer's concerns on the focus of the manuscript and the choice of materials in the experimental section.

Regarding the focus of this paper, our primary aim was to develop a model that obeys detailed balance and can relate phase behavior to the static disorder. We felt that the model would be more valuable when applied to explain relevant experimental systems. It is true that the selected P3HT:NFA systems show non-trivial phase behaviour, but we considered it useful to connect the model to materials that do exhibit static disorder and whose phase behaviour has already been studied in some detail. We agree that the different trends shown by the two materials systems increase the amount of information in the paper. On the other hand, the contrasting phase behaviour, and resultant CT state behaviour, that result from a small change in chemical structure of the NFA is likely to be interesting to the field, particularly to materials chemists, and in our view more useful than considering only one blend system, or considering different blend systems that have no connection in their chemical structure.

Therefore, the discussion on the relationship and connection between phase behavior and static disorder is necessary in the main text, i.e. section "**Relating phase behaviour to static disorder**". However, we agree that in the modelling part, i.e. Section "**Reproducing the experimental trends using the static disorder model**", the simulations of O-IDTBR is less useful to test our model as we didn't observe changes of energetic distribution of CT states as a function of NFA content, we therefore have moved the simulations of O-IDTBR into Supplementary Note 3 in the SI (see also the responses to **Comment 29**).

Comment 35:

35) The sentence "It is therefore important to account for disorder in CT state energy before deriving any material related trends or design rules from analysis of experimental data." appears somewhat unrelated to the MS because it does not aim at any design rules.

Responses:

We apologize for the confusion caused. To make it clearer, we have therefore extended the discussion in the discussion section to provide our understanding of our results and some guidelines for low static disorder material design (see also below the changes in blue):

“This work presents a model that can be used to quantify and understand the impact of disorder in charge-transfer state energy (static disorder) on voltage losses in organic photovoltaic devices. We demonstrated that static disorder tends to increase voltage losses in OPV devices, as previously reported, and that it is indeed important in certain well-studied polymer: molecular acceptor material systems, in agreement with Ref. ^{9,11,17}. In the literature it has been common to analyse external quantum efficiency and emission data in terms of a single CT state. We find that, when spectra from a system which possesses static disorder are interpreted in terms of a single state energy, the analysis yields incorrect values of CT state energy, the associated reorganization energy and voltage losses. It is therefore important to account for disorder in CT state energy before deriving any material related trends or guidelines from analysis of experimental data. Several recent studies reporting the relative insignificance of static disorder in CT state energies in organic blends focussed on dilute dispersion of donor molecules in a fullerene matrix. ^{14,16,18} Such material systems have been widely studied precisely because morphological disorder is minimised. In contrast, the conjugated polymer:molecular acceptor blends commonly used for high performance OPV bring several sources of energetic disorder not found in the dilute small molecule blends, namely, the large conformational – and therefore energetic – phase space of conjugated polymers, the electronic anisotropy in both components and the sizeable volume fractions that permit aggregation of both components. In these more widely studied materials, understanding the relationship between phase behaviour and static disorder is important.

Our study explored the relationship between phase behaviour and static disorder in two polymer:small molecule blends, P3HT:O-IDTBR and P3HT:O-IDFBR, where we modulated the interface properties through structure of the small molecule and blend composition. In both cases, different CT state features appear that are associated with amorphous and crystalline phases of the materials, demonstrating the impact of

structural heterogeneity on CT-state energy disorder^{21,29,32,33}. Different material phases within the same system tend to enhance the negative effects of CT state energy disorder. The P3HT:O-IDFBR experimental system helps to validate a key finding from a theoretical model of static disorder, which is that increasing static disorder is generally detrimental to voltage losses (reducing the radiative open circuit voltage and increasing the non-radiative voltage loss). Given that a sharp onset to the CT-state DoS would help to minimise static disorder, then a blend of two crystalline components may be preferred. However the lower energy of states in crystalline domains tends to compromise voltage losses via the energy gap law^{7,8}. Amorphous components may reduce non-radiative voltage losses that result from high CT-state energy but will generally bring disorder in CT state energies and will ultimately be compromised by transport limitations. The combination of a low band-gap, sharp-onset, crystalline molecule with a less crystalline polymer that presents a low interfacial energy offset would help to minimise static disorder whilst maintaining electronic connectivity. These features are present in the high-performance PM6:Y6 blend, which shows evidence of a lower degree of static disorder both in the shape of its EQE onset and in its temperature dependence than the blends studied^{67,68}.

Our study demonstrates the importance of accounting for CT-state energy disorder in analysing the voltage losses in OPV devices. Ultimately, approaching the limits to performance will require such static disorder be minimised. In polymer:molecule blends, this will mean reducing the disorder that arises from the conformational phase space of polymers, from aggregation and from molecular anisotropy. At the same time, good electronic and excitonic transport must be maintained. The task will require more detailed experimental^{69,70} and theoretical⁷¹ probes of molecular geometry and packing arrangement at interfaces and its relationship with CT-state energy.”

Comment 36:

36) *It is not clear what is meant by the conclusion “Our combined experimental and modelling study demonstrates how the model can be used to explore the relationship between phase behaviour and static disorder in a blend.”, because the relationship between phase behaviour (not studied in the MS) and static disorder has not been studied but has been suggested by the authors.*

Responses:

We apologize for the confusion caused while reading the original draft. In the revised experimental section, we believe it’s clear that the relationship between phase behaviour and static disorder is investigated. Please refer to Section “**Relating phase behaviour to static disorder.**” in the main text for detailed discussions on how phase behaviour can be related to static disorder using EQE, EL data with the help of the detailed phase behaviour study in Ref. Chem. Mater. 32, 8294–8305 (2020). We have also summarized the key findings in the manuscript and provided some guidelines for low static disorder materials design (see also the response to **Comment 35**).

Comment 37:

37) *The phrase “experimental validation of the model” in the abstract is questionable in view of the specific input needed to reproduce the experiments. It appears that the model has been tailored to reproduce some experiments.*

Response:

We have changed this sentence in the abstract to:

“Finally, we demonstrate the importance of the model by showing that the experimental absorption, emission, and voltage loss behaviour in a series of polymer:nonfullerene blends of varying composition and microstructure can only be explained when static disorder is considered.”

We have also modified the last paragraph of introduction to:

“In this contribution, we introduce a model that incorporates static disorder in CT-state energy by including a general distribution of

electronic CT states $g(E_{CT})$. Using our model, we first show that the principle of detailed balance is obeyed regardless of the shape of $g(E_{CT})$, and that use of single-CT-state analysis to quantify emission and absorption can lead to incorrect results for voltage losses when static disorder is not considered. We test our model using two series of poly(3-hexylthiophene-2,5-diyl) (P3HT): non-fullerene acceptors (NFAs) blends as a function of NFA content, in which we observe the presence of two distinct CT-state features that we assign to the presence of semi-crystalline and amorphous phases of P3HT. We demonstrate that using our model, we can successfully reproduce the observed experimental changes in the absorption and emission spectra as well as in the voltage behaviour of the devices.”

Report of Reviewer #3:

Reviewer's overall evaluation:

The authors of this manuscript suggest that static disorder is the dominant contribution and present a physical model to calculate the influence of disorder on voltage (and, thus, efficiency) losses in OPVs. This model is applied to two prototypical polymer:non-fullerene blends showing different impact of their mixing ratio on the interfacial donor-acceptor morphology and, thus, the CT DOS. Model predictions and experimental results are compared to obtain valuable insights in voltage losses of these devices.

Overall, the manuscript is sound and represents a significant step forward to better understanding OPV devices. It can be published after some issues have been addressed:

Responses:

We really appreciate **Reviewer #3**'s positive comments on our manuscript. We also thank **Reviewer #3** for the helpful suggestions that led us to improve our manuscript.

Comment 1:

1. Before presenting their own model, the authors list previous work and the related shortcomings in the context of properly including static disorder effects (p. 4). Specifically, they pick the model by Kahle et al. (Ref. 11) to demonstrate that it does not fulfil detailed balance (Fig. S1). However, it is not specified which of the different expressions given in Ref. 11 is actually used for the simulation; let alone the required input parameter values to produce the curves shown in Fig. S1. Moreover, the authors should add some discussion in the SI as to why this expression does not fulfil DB. I think it should even be possible to show this analytically (in addition to the simulation).

Responses:

We thank **Reviewer #3** for this nice suggestion. Following this comment, we have listed the equations and relevant parameter values used to produce Supplementary Fig. 1 as well as the analytical approach in the SI. We have also added some text in the SI, as also shown below (in blue):

Supplementary Note 1. Equations for modelling Supplementary Fig. 1

The equations used to perform the simulation in Supplementary Fig. 1 are the high temperature limit of Equation (8a-b) in Ref. ¹⁸ (i.e. Kahle model), which is also quoted as Equation (9) and Equation (10) in Ref. ¹⁹, as shown here:

$$rk_r(\hbar\omega) = \frac{k_r(\hbar\omega)}{\hbar\omega} \propto \exp\left\{-\frac{[\hbar\omega - (E_{CT} - \lambda_o)]^2}{4\lambda_o k_B T + 2\sigma^2}\right\}, \quad (\text{N1.1a})$$

$$rA(\hbar\omega) = A(\hbar\omega) \cdot \hbar\omega \propto \exp\left\{-\frac{[\hbar\omega - (E_{CT} + \lambda_o)]^2}{4\lambda_o k_B T + 2\sigma^2}\right\}, \quad (\text{N1.1b})$$

Where $rk_r(\hbar\omega)$ and $rA(\hbar\omega)$ are the reduced rate of emission and reduced absorptance. Note here we use the same notation for emission and absorptance as in our model to avoid confusion. In the simulation, the parameters were chosen as: $E_{CT} = 1.75 \text{ eV}$, $\lambda_o = 0.2 \text{ eV}$, and $\sigma = 0.1 \text{ eV}$.

Here, we also use an analytical method to check the validation of detailed balance of the Kahle model. In order to obey detailed balance, Supplementary Equation (M5.1) must be valid. To make it clearer analytically, we here get rid of the energy term ($\hbar\omega$) except for the exponential term in Supplementary Equation (M5.1), then we have:

$$\frac{rk_r(\hbar\omega)}{rA(\hbar\omega)} \propto \exp\left(-\frac{\hbar\omega}{k_B T}\right), \quad (\text{N1.2})$$

However, when we perform the division using Supplementary Equation (N1.1) we have:

$$\frac{rk_r(\hbar\omega)}{rA(\hbar\omega)} \propto \frac{\exp\left\{-\frac{[\hbar\omega - (E_{CT} - \lambda_o)]^2}{4\lambda_o k_B T + 2\sigma^2}\right\}}{\exp\left\{-\frac{[\hbar\omega - (E_{CT} + \lambda_o)]^2}{4\lambda_o k_B T + 2\sigma^2}\right\}} = \exp\left\{-\frac{4(\hbar\omega - E_{CT})\lambda_o}{4\lambda_o k_B T + 2\sigma^2}\right\}$$

$$\propto \exp\left\{-\frac{4\hbar\omega\lambda_o}{4\lambda_o k_B T + 2\sigma^2}\right\} \neq \exp\left(-\frac{\hbar\omega}{k_B T}\right), \text{ (N1.3)}$$

Therefore, the principle of detailed balance cannot be satisfied using Equations N1 when σ is non-zero.

Comment 2:

2. The new model is based in part on work published earlier in 2018 (Ref. 8). Nevertheless, the authors should try to present the theory in a way that readers get the basic idea without having to dig too deep in the literature. This holds especially for the presentation on p. 6 of the manuscript. The expressions for the FCWD – by the way, the acronym needs to be defined as well – appear all by a sudden without any introduction how they are to be used. I think it would be much easier to follow if parts of section 3 from the SI are transferred to the main manuscript text.

Responses:

Following this suggestion, we have added some text on the background of this model, where the key expressions have also been listed in the main text. The revised version of the model is now very detailed in the main text (see also below for the changes in blue):

“Static disorder model” section in the main text:

“Static disorder model. Fig. 1a-b illustrates a general energetic distribution of CT states, $g(E_{CT})$, that might result at a D-A heterointerface in which a donor domain is surrounded by acceptor domains of different sizes and strength of interaction with the donor²⁸. In general, such a distribution is likely to contain a number of CT-state manifolds, each centred at $E_{CT,t}$, where t denotes the order of CT manifold. We thus describe the static disorder in the system by a normalised distribution function for CT-state energies given by

$$g(E_{CT}) = \sum_t c_t D_t(E_{CT}), \text{ (1)}$$

where $D_t(E_{CT})$ is a line-shape function and c_t is a weighting coefficient such that $\sum_t c_t = 1$. Practically, $D_t(E_{CT})$ can be any

function. In our model we use a gaussian line-shape for $D_t(E_{CT})$, i.e.

$$D_t(E_{CT}) = \frac{1}{\sigma_{CT,t}\sqrt{2\pi}} \exp \left[-\frac{1}{2} \left(\frac{E_{CT} - E_{CT,ct}}{\sigma_{CT,t}} \right)^2 \right],$$

where $\sigma_{CT,t}$ is the width of the individual gaussian function. In the calculations shown below, each gaussian function is integrated in the range of $[E_{CT,ct} - 5\sigma_{CT,t}, E_{CT,ct} + 5\sigma_{CT,t}]$, ensuring $\int_a^b D_t(E_{CT})d(E_{CT}) = 1$, hence $\int g(E_{CT})d(E_{CT}) = 1$, where $a = E_{CT,ct} - 5\sigma_{CT,t}$ and $b = E_{CT,ct} + 5\sigma_{CT,t}$.

As in previous models^{7,8,38}, we assume that radiative and non-radiative recombination in the device follow the energy gap law^{43,44}, and occur only via the CT states, either directly after exciton dissociation or by reformation of the CT state from free charges. We additionally assume that absorption, emission and non-radiative recombination transitions occur in the weak coupling limit so that they can be described by non-adiabatic Marcus theory, and we invoke the Franck Condon principle to include transitions between different vibrational modes, as illustrated in Fig. 1c. Here, we adopt the method introduced by Jortner to describe the rate of transition between the CT state and the ground state⁴⁵, in which we distinguish between high and low frequency vibronic modes. We consider that the states in each CT state manifold (denoted as t in the subscript) share the same set of parameters, such as oscillator strength ($f_{osc,t}$) and low frequency reorganization energies ($\lambda_{o,t}$), except for their energies (E_{CT}). The total rate of non-radiative recombination from the CT state can be expressed as the sum of the contribution from all CT manifolds, as:

$$K_{nr} = \sum_t \int_a^b \frac{2\pi}{\hbar} V_t(E_{CT})^2 FCWD_{rec,t}(0, E_{CT}) c_t D_t(E_{CT}) d(E_{CT}), \quad (2)$$

where $V_t(E_{CT})$ is the electronic coupling between CT and ground state described by the generalized Mulliken-Hush method^{46,47}. $FCWD_{rec,t}$ is the Franck-Condon-Weighted Density of States for recombination for CT state manifold t . The radiative recombination and absorption rates

can be expressed as the sum of the contribution from all CT manifolds using a similar expression and all depend on the Franck-Condon-Weighted Density of States (FCWD)⁸, via

$$k_{abs}(\hbar\omega) = \sum_t \int_a^b \frac{W}{3\pi\epsilon_0\hbar^4} \left(\frac{\hbar\omega}{c}\right)^3 M_t(E_{CT})^2 FCWD_{abs,t}(\hbar\omega, E_{CT}) c_t D_t(E_{CT}) d(E_{CT}), \quad (3a)$$

$$k_r(\hbar\omega) = \sum_t \int_a^b \frac{1}{3\pi\epsilon_0\hbar^4} \left(\frac{\hbar\omega}{c}\right)^3 M_t(E_{CT})^2 FCWD_{rec,t}(\hbar\omega, E_{CT}) c_t D_t(E_{CT}) d(E_{CT}), \quad (3b)$$

where W is the photon density and accounts for the strength of electromagnetic field around the molecule;⁴⁸ ϵ_0 is the permittivity of the free space; $M_t(E_{CT})$ is the transition dipole moment for CT manifold t and is related to the oscillator strength of the CT manifold ($f_{osc,t}$) under the dipole approximation^{49,50} via $M_t(E_{CT}) = \sqrt{(3/2)q^2\hbar^2 f_{osc,t}/(E_{CT}m_e)}$, with m_e the electron mass and q the elementary charge. K_r (s^{-1}) is obtained by integrating Equation (3b) over $\hbar\omega$, via

$$K_r = \int_0^{\infty} k_r(\hbar\omega) d\hbar\omega. \quad (4)$$

We assume quasi-thermal equilibrium (QTE) conditions, meaning that the occupation function of each electronic CT state should be considered in the expression for photon emission, and that state occupation should follow Boltzmann statistics under low injection conditions. This leads to modified expressions from MLJ theory^{7,8} for FCWD for absorption ($FCWD_{abs,t}(\hbar\omega, E_{CT})$) and recombination ($FCWD_{rec,t}(\hbar\omega, E_{CT})$) for CT state manifold (t),

$$\begin{aligned} & FCWD_{abs,t}(\hbar\omega, E_{CT}) \\ &= \frac{1}{Z_{abs}} \frac{1}{\sqrt{4\pi\lambda_{o,t}k_B T}} \\ &\times \sum_{m=0}^{\infty} \sum_{n=0}^{\infty} \frac{e^{-S_t} S_t^{n-m} m!}{n!} [(L_m^{n-m}(S_t))]^2 \exp\left[-\frac{(-\hbar\omega + E_{CT} + \lambda_{o,t} + (n-m)\hbar\Omega_t)^2}{4\lambda_{o,t}k_B T}\right] \exp\left(-\frac{m\hbar\Omega_t}{k_B T}\right), \quad (5a) \end{aligned}$$

$$\begin{aligned}
& FCWD_{rec,t}(\hbar\omega, E_{CT}) \\
&= \frac{1}{Z_{rec}} \frac{1}{\sqrt{4\pi\lambda_{o,t}k_B T}} \\
&\times \sum_{m=0}^{\infty} \sum_{n=0}^{\infty} \frac{e^{-S_t} S_t^{n-m} m!}{n!} [(L_m^{n-m}(S_t))]^2 \exp\left[-\frac{(\hbar\omega - E_{CT} + \lambda_{o,t} + (n-m)\hbar\Omega_t)^2}{4\lambda_{o,t}k_B T}\right] \exp\left(-\frac{m\hbar\Omega_t + E_{CT} - \mu}{k_B T}\right), \quad (5b)
\end{aligned}$$

where $\lambda_{o,t}$ and $\lambda_{i,t}$ are the low frequency and high frequency reorganization energy for CT manifold t , respectively, $S_t = \lambda_{i,t} / \hbar\Omega_t$ is the Huang Rhys factor⁵¹, $\hbar\Omega_t$ is the averaged harmonic energy spacing, typically 0.15-0.20 eV for molecules made of many carbon-carbon bonds^{7,8} and m and n are the quantum numbers of the vibrational mode of the initial and final state, respectively. $L_m^{n-m}(S_t)$ is then the generalized Laguerre polynomial of degree m ⁸. The constraint of equilibrated occupation of electronic states following Boltzmann statistics ensures that the different D-A interfaces in the ensemble share the same chemical potential energy (μ) and the system is in QTE. μ is defined as the free energy gained by the system under injection or illumination conditions, and is also known as the Fermi energy in inorganic semiconductors. The factor Z_{abs} and Z_{rec} in Equation (5) are the partition functions, and are defined as the sum of occupation of vibrational CT states for absorption, and as the sum of occupation of both vibrational and electronic CT states for recombination, and are given by

$$Z_{abs} = \sum_t \sum_{m=0}^{\infty} \exp\left(-\frac{m\hbar\Omega_t}{k_B T}\right), \quad (6a)$$

$$Z_{rec} = \sum_t \left\{ \sum_{m=0}^{\infty} \exp\left(-\frac{m\hbar\Omega_t}{k_B T}\right) \int_a^b c_t D_t(E_{CT}) \exp\left(-\frac{E_{CT} - \mu}{k_B T}\right) d(E_{CT}) \right\}. \quad (6b)$$

When the absorption and emission spectra are constructed from these FCWD distributions, the spectra represent the sum of contributions from different CT state energies, and the resulting absorption and emission bands are broadened not only by the low frequency

broadening of each vibronic line and the FC vibronic progression, but also by the distribution of CT state energies. Attempts to interpret such spectra in terms of a single CT state energy would lead to incorrect estimates for both the CT state energy and the reorganisation energies. We also remark in passing that in the case where the whole system is not in QTE, for example if the spatially separated states are not strongly coupled, as explored by Melianas et al⁵², we would, in general, expect different chemical potential energies at different manifolds (sites) of CT states, and the above expression for emission would need to be adapted. The full model describing the voltage losses as well as the rates of the radiative and non-radiative transitions with static disorder included is presented in Supplementary Method 2-4.”

Comment 3:

3. Furthermore, I find it hard to compare simulation results if the input parameters are not fully specified (see e.g. Fig. 3 or Fig. S3). Often, only the varied parameter values are given (Fig. S4-S8) but the rest is unclear. I suggest to list all input parameter, e.g. in another table in the SI.

Responses:

We thank the reviewer’s suggestion. We have added text to the caption for Figure 3 and Figure S3 to state that the parameters used are given in Supplementary Table 1. The new text added to the figure caption for each figure is shown below in blue.

Figure caption of Figure 3:

“Fig. 3. Single-state analysis on simulated spectrum. a Example of single-state analysis using reduced emission ($\propto k_r/(\hbar\omega)^3$) and absorption ($\propto A/\hbar\omega$) spectrum. b Extracted $E_{CT,eff}$ and λ_{eff} as a function of σ_{CT} using 1-gaussian DoS as an example. Simulated emission and absorption spectrum are used as the input for single-state analysis. Details of the comparisons on the calculated emission, absorption, recombination rate, and voltage losses are shown in

Supplementary Fig. 11. The parameters used to produce the input spectrum are listed in Supplementary Table 1.”

Figure caption of Supplementary Fig. 3:

“**Supplementary Fig. 3. Extra nonradiative voltage losses caused by static disorder using Burke model²² as compared to the results from our model.** In both models, we use one gaussian density of state of CT state. We note that energy gap law is not considered in Burke model, and the extra nonradiative voltage loss follows $\sigma_{CT}^2/2k_B T$. We see that Burke model significantly overestimates the extra nonradiative voltage losses as compared to our mode results. The parameters used to calculate the nonradiative voltage losses using our model are listed in Supplementary Table 1.”

Regrading Supplementary Fig. 4-8, indeed there is a need to better present the parameters used in the simulations, as they are complicated simulations. To facilitate this, we added a new table (named Supplementary Table 4 in the SI), see also here:

Supplementary Table 4. Key parameter values used in **Supplementary Fig. 4-8**

Parameter	Value when it's constant	Value when it's varied	Units
Difference in the static dipole moment ($ \overline{\Delta\mu} $)	5	N/A	Debye
Oscillator strength (f_{osc})	1×10^{-2}	$[10^{-2}, 1]$	Unitless
High frequency reorganisation energy (λ_i)	0.2	$[0.05, 0.2]$	eV
Low frequency reorganisation energy (λ_o)	0.2	$[0.05, 0.2]$	eV
Vibrational mode harmonic oscillator energy ($\hbar\Omega$)	0.15	N/A	eV
refractive index (n)	1.5	N/A	Unitless
The ratio of CT state density ($N_{CT/LE}$)	1	$[10^{-3}, 1]$	Unitless
Energy of local excitonic state (E_{LE})	1.8	N/A	eV
Energy of the centre of CT states ($E_{CT,C}$)	1.5	$[1.5, 1.65]$	eV

$g(E_{CT})$ Gaussian width (σ_{CT})	0 or 0.1	N/A	eV
--	----------	-----	----

Note: the values in square brackets give the range of changing parameter values. To produce Supplementary Fig. 4-8, when we perform simulations on one varied parameter, for example, $E_{CT,C}$, we assume other parameter are constant, which are listed in the second column (“Value when it’s constant”). Light yellow shaded area indicates the variable(s).

In addition to the figures that the reviewer mentioned, we have also checked other figures to make sure the input parameters have been clearly listed and explained.

We also note here that two new tables in the SI, i.e. Supplementary Table 2 and Supplementary Table 3, have been added following the revised Figure 2.

Comment 4:

4. The application of the model to experimental data is performed in a qualitative manner without fitting. Nevertheless, it seems to yield valid results. Just two comments: (i) While temperature dependent EQE indicates that static disorder is dominant (Fig. S15), there is no information about T-dependent EL spectra. (ii) Is it possible to label the energy axis of Fig. S18?

Responses:

We thank the reviewer for this comment.

We actually performed both temperature dependent EQE and EL measurements. Whilst we are confident in the quality of the T-dependent EQE spectra, we have lower confidence in the T-dependent EL spectra and could not obtain reliable spectra for some of the devices at low temperature. Under low current injection, the luminescence was too weak whilst under high injection the devices were easily shorted at low temperature. This problem prevented us from obtaining useful low temperature EL for O-IDTBR devices. We did manage to collect temperature-dependent CT-EL data from O-IDFBR devices with different NFA wt% from 20% to 70%. The spectra at 300K and 80K are shown in Supplementary Figure 16 (see also below). Although the data is not very ideal, it’s clear that there is no clear evidence of sharpening of EL emission spectrum in consistent with the T-dependent EQE results (see also the reproduced EQE using EL, via EL/Φ_{BB}) when temperature is varied in a large range, supporting the conclusion drawn using injection-dependent EL and temperature dependent EQE that static disorder is significant in the devices we studied here.

We have also added Supplementary Figure 16 in the SI, and revised the text in line 295-299 in the main text to:

“Further evidence of static disorder for the studied devices has also been obtained in T-dependent EQE and EL measurements, where we observed no sharpening of the EQE or EL tail with reduced temperature as shown in Supplementary Fig. 15 and Supplementary Fig. 16. This is consistent with a picture in which static disorder is dominant over dynamic disorder¹¹.”

We note here that we have moved the evidence of static disorder by temperature dependent EQE and EL up to the beginning when we made that point (around the discussion of room temperature EQE), which offers a better flow for the paper.

Supplementary Figure 16. Temperature dependent EL and EQE for O-IDFBR devices with different NFA wt% from 20% to 70%. (a-c) EL; (d-f) Experimental EQE and reproduced EQE using EL. Devices are measured at two different temperatures, i.e. 300K and 80K. The injection current for 20%, 40%, and 70% O-IDFBR devices are 70 mA, 50 mA and 10 mA, respectively. Reproduced EQE spectra using EL spectra based on the reciprocity relation are show in (d-f) using solid lines. It’s clear that there is no clear evidence of sharpening of EL emission spectrum in consistent with the T-dependent EQE results (see also the

reproduced EQE using EL, via EL/Φ_{BB}) when temperature is varied in a large range, supporting the conclusion drawn using injection-dependent EL and temperature dependent EQE that static disorder is significant in the devices we studied here. We note here that for O-IDTBR devices, we didn't manage to acquire reliable T-dependent EL data.

Regarding point (ii), the energy axis of Supplementary Figure 19 has been labeled (see below):

Supplementary Figure 19. Illustration of estimated distribution of $g(E_{CT})$ as a function of composition.

Comment 5:

5. There are some typos that should be corrected:

- Fig. 1, caption: “triple” should read “triplet”

- Fig. 6, caption: Where in the graphs do I find “F_C1” and “F_C2”?

- SI, section 1: Lockin is probably an SRS830 (and not 380) model. Amplification probably 10^6 V/A (and not 106)!

- Fig. S11-13: Legend should read “Dashed”.

- Fig. S11: Caption should refer to Fig. 3a (and not 2a) of main text.

Responses:

Thank you for pointing out those typos. They have all been corrected.

We have also revised Fig. 6 to improve the readability in the main text. Since F_{C2} wasn't varied in the simulations, it has been removed from the figure caption of Fig. 6.

We sincerely hope that **Reviewer #1, Reviewer #2 and Reviewer #3** are satisfied with our detailed responses, and that these changes made to the text have allowed us to assemble an improved and stronger manuscript.

REVIEWER COMMENTS

Reviewer #1 (Remarks to the Author):

Authors have made substantial changes to the manuscript which have clearly improved clarity, also for non-expert readers. My previous comments have been addressed in this revised version and I still recommend publication.

Reviewer #2 (Remarks to the Author):

The manuscript by Yan et al. reports a comprehensive theoretical and experimental work on charge-transfer (CT) states in organic solar cell blends. It starts with simulations of blends based on static disorder models and analyzes the effect of disorder on decay rates of CT states and voltage losses in OPV cells. Going beyond a single Gaussian functions in the density of CT states, the authors use a series of Gaussians in the model. The influence of this extension of the model on radiative and non-radiative rates are studied. In the second part of the manuscript, they perform experiments for P3HT:NFA blends with two different NFAs, including absorption (EQE) and electroluminescence. The manuscript has improved over the previous version for which I listed a number of issues. Most points have been adequately addressed and I think one should follow the track towards publication in Nature Communications because the work should be of interest to its readership. Although the new version of the manuscript is clearer in many details, there are a few points to be solved.

I still feel that the different parts of the manuscript

(a) the theoretical study in the beginning

(b) the comparison of simulations with disorder and single-state fitting of experimental data

(c) the study of the NFA-based blends with its dedicated modelling

appear to be “not naturally connected” and have their own specific identities.

I acknowledge the author’s motivation “We felt that the model would be more valuable when applied to explain relevant experimental systems.” However, regarding the structure, it would be probably clearer to say from the beginning that they study two cases with relevant disorder: one with the models (including above parts (a) and (b)) and another case with the NFA-blends which is more specific with two distinct CT state manifolds.

The discussion of the chemical potential (μ) is still questionable. Within thermodynamics, the concept of the chemical potential is usually related to a specific species. The discussion around this point is confusing. Is μ introduced for electrons, holes or both? In fact, in Eq. 5b of the model, μ actually cancels (see Eq. 6b), which is not surprising for Boltzmann statistics. Therefore, any dependence of results on μ must be an artifact. My feeling is that the authors don’t really need this concept at that point. Later on there are simulations with state filling effects for which the authors switch to a different model with Fermi-Dirac distribution.

On page 5: What means they assume that radiative and non-radiative recombinations follow the energy gap law? How is this additional assumption included?

Why are the curves in Fig. 2a appear to be so wobbly? Is it because of the small linewidth? Please indicate the value of $\sigma_{CT,t}$ or λ that is causing this effect in the caption.

The authors talk about single Gaussian CT state distribution. Please clarify that you mean that the coefficients c_t are distributed like a Gaussian (if this is what is meant) and not that the resulting distribution of states is a Gaussian.

Reviewer #3 (Remarks to the Author):

The authors have done an impressive job in responding to the reviewers' comments and revising their manuscript accordingly. It can now be accepted for publication without further changes.

Point-by-point Response to the Reviewers' Comments

Manuscript NCOMMS-20-48130A

We thank the reviewers for their comments and constructive suggestions. Please find below the point-by-point responses to the comments. To ease readability, we are repeating the reviewers' comments in *italics*.

Reviewer(s)' Comments to Author:

Reviewer #1 (Remarks to the Author):

Authors have made substantial changes to the manuscript which have clearly improved clarity, also for non-expert readers. My previous comments have been addressed in this revised version and I still recommend publication.

Reviewer #3 (Remarks to the Author):

The authors have done an impressive job in responding to the reviewers' comments and revising their manuscript accordingly. It can now be accepted for publication without further changes.

Responses:

We appreciate that **Reviewer #1** and **Reviewer #3** are satisfied with our response to the first-round comments and that they both are happy to accept the manuscript.

Reviewer #2 (Remarks to the Author):

Reviewer's overall evaluation:

The manuscript by Yan et al. reports a comprehensive theoretical and experimental work on charge-transfer (CT) states in organic solar cell blends. It starts with simulations of blends based on static disorder models and analyzes the effect of disorder on decay rates of CT states and voltage losses in OPV cells. Going beyond a single Gaussian functions in the density of CT states, the authors use a series of Gaussians in the model. The influence of this extension of the model on radiative and non-radiative rates are studied. In the second part of the manuscript, they perform experiments for P3HT:NFA blends with two different NFAs, including absorption (EQE) and electroluminescence.

The manuscript has improved over the previous version for which I listed a number of issues. Most points have been adequately addressed and I think one should follow the track towards publication in Nature Communications because the work should be of interest to its readership. Although the new version of the manuscript is clearer in many details, there are a few points to be solved.

Responses:

We very much appreciate that **Reviewer #2** believes that our manuscript has been improved and has provided additional comments to further strengthen the paper. We have carefully addressed her/his additional concerns as follow and made revisions in the manuscript accordingly.

Comment #1:

I still feel that the different parts of the manuscript

(a) the theoretical study in the beginning

(b) the comparison of simulations with disorder and single-state fitting of experimental data

(c) the study of the NFA-based blends with its dedicated modelling

appear to be “not naturally connected” and have their own specific identities.

I acknowledge the author's motivation "We felt that the model would be more valuable when applied to explain relevant experimental systems." However, regarding the structure, it would be probably clearer to say from the beginning that they study two cases with relevant disorder: one with the models (including above parts (a) and (b)) and another case with the NFA-blends which is more specific with two distinct CT state manifolds.

Responses:

Following this suggestion, we have added a short paragraph at the end of "Static Disorder Model" Section to clarify the structures of the paper (see also below in blue):

"In the following sections, we first investigate the impact of static disorder on emission, absorption, and voltage losses analysis from a theoretical point of view, using the model developed here. We then apply the model to investigate two experimental P3HT:NFA systems, where static disorder is evident from the CT state emission spectra and can be related to the phase behaviour of the blends."

Comment #2:

The discussion of the chemical potential (μ) is still questionable. Within thermodynamics, the concept of the chemical potential is usually related to a specific species. The discussion around this point is confusing. Is μ introduced for electrons, holes or both? In fact, in Eq. 5b of the model, μ actually cancels (see Eq. 6b), which is not surprising for Boltzmann statistics. Therefore, any dependence of results on μ must be an artifact. My feeling is that the authors don't really need this concept at that point. Later on there are simulations with state filling effects for which the authors switch to a different model with Fermi-Dirac distribution.

Responses:

We appreciate the reviewer raises this question. Here, we would like to address the issue in detail to avoid further confusion. We also note here that the changes made here only serve to make the manuscript more accessible, and do not affect the results and analysis in the manuscript.

The chemical potential energy (μ) is related to the charge transfer (CT) states, defining the state occupation for CT states under injection condition (light or bias) following Boltzmann statistics. This point has been made clearer in page 8:

“ μ is defined as the free energy of CT state excitons in the biased system and quantifies the disturbance from equilibrium (where $\mu = 0$).”

The reviewer is right that the chemical potential cancels out in equations 5 and 6 in previous manuscript given the system is under quasi-thermal-equilibrium (QTE) condition. The reason it was presented like that was to enable the exploration of the effect of a non-uniform μ such as that was discussed in Supplementary Note 5, which is an interesting point recently pointed out by Melianas et al. (see Proc. Natl. Acad. Sci. U. S. A. 116, 23416–23425 (2019)). In that case, the chemical potentials do not cancel out. But we can see that the presentation of chemical potential in the previous manuscript was indeed confusing, and it is especially confusing at the place where the injection dependence of EL is modelled later. So, we have tried to fix these confusing aspects as follows:

- 1) We first removed the chemical potential (μ) from Eq. (5) and Eq. (6) and presented the equations in the theory part in a clearer manner by presenting the Boltzmann term and Z factors in rate constants equations, see changes in the “Static disorder Model” section in blue. We also copy the new equations here:

$$K_{nr} = \frac{1}{Z_{rec}} \sum_t \int_a^b \frac{2\pi}{\hbar} V_t(E_{CT})^2 FCWD_{rec,t}(0, E_{CT}) c_t D_t(E_{CT}) \exp\left(-\frac{E_{CT}}{k_B T}\right) d(E_{CT}), \quad (2)$$

$$k_{abs}(\hbar\omega) = \frac{1}{Z_{abs}} \sum_t \int_a^b \frac{W}{3\pi\epsilon_0\hbar^4} \left(\frac{\hbar\omega}{c}\right)^3 M_t(E_{CT})^2 FCWD_{abs,t}(\hbar\omega, E_{CT}) c_t D_t(E_{CT}) d(E_{CT}), \quad (3a)$$

$$k_r(\hbar\omega) = \frac{1}{Z_{rec}} \sum_t \int_a^b \frac{1}{3\pi\epsilon_0\hbar^4} \left(\frac{\hbar\omega}{c}\right)^3 M_t(E_{CT})^2 FCWD_{rec,t}(\hbar\omega, E_{CT}) c_t D_t(E_{CT}) \exp\left(-\frac{E_{CT}}{k_B T}\right) d(E_{CT}), \quad (3b)$$

$$FCWD_{abs,t}(\hbar\omega, E_{CT}) = \frac{1}{\sqrt{4\pi\lambda_{o,t}k_B T}} \times \sum_{m=0}^{\infty} \sum_{n=0}^{\infty} \frac{e^{-S_t} S_t^{n-m} m!}{n!} [L_m^{n-m}(S_t)]^2 \exp\left[-\frac{(-\hbar\omega + E_{CT} + \lambda_{o,t} + (n-m)\hbar\Omega_t)^2}{4\lambda_{o,t}k_B T}\right] \exp\left(-\frac{m\hbar\Omega_t}{k_B T}\right), \quad (5a)$$

$$\begin{aligned}
& FCWD_{rec,t}(\hbar\omega, E_{CT}) \\
&= \frac{1}{\sqrt{4\pi\lambda_{o,t}k_B T}} \\
&\times \sum_{m=0}^{\infty} \sum_{n=0}^{\infty} \frac{e^{-S_t} S_t^{n-m} m!}{n!} [(L_m^{n-m}(S_t))]^2 \exp\left[-\frac{(\hbar\omega - E_{CT} + \lambda_{o,t} + (n-m)\hbar\Omega_t)^2}{4\lambda_{o,t}k_B T}\right] \exp\left(-\frac{m\hbar\Omega_t}{k_B T}\right), \quad (5b)
\end{aligned}$$

$$Z_{rec} = \sum_t \left\{ \sum_{m=0}^{\infty} \exp\left(-\frac{m\hbar\Omega_t}{k_B T}\right) \int_a^b c_t D_t(E_{CT}) \exp\left(-\frac{E_{CT}}{k_B T}\right) d(E_{CT}) \right\}. \quad (6b)$$

- 2) Secondly, we presented new equations of recombination rate (R_r, R_{nr}) (not rate constant) to account for the effect of chemical potential on the recombination rate magnitude under QTE condition in page 8, and we also made it clearer that under QTE condition we should have a uniform chemical potential and the emission spectra do not change with chemical potential, see also here in blue:

“In QTE, the volumetric radiative (R_r) and non-radiative (R_{nr}) recombination rates of the system when under optical or electrical bias are controlled by the chemical potential energy, μ , of the system. μ is defined as the free energy of CT state excitons in the biased system and quantifies the disturbance from equilibrium (where $\mu = 0$). In QTE all CT states in the ensemble share the same chemical potential energy (μ) and R_r, R_{nr} are amplified from their equilibrium values by the factor $\exp\left(\frac{\mu}{k_B T}\right)$ such that

$$R_{nr} = K_{nr} n_{CT0} * \exp\left(\frac{\mu}{k_B T}\right), \quad (7a)$$

$$R_r = K_r n_{CT0} * \exp\left(\frac{\mu}{k_B T}\right), \quad (7b)$$

where n_{CT0} is the density of CT states in thermal equilibrium.

In QTE, the recombination rate constants (K_r, K_{nr}) and emission spectra shape (k_r) do not change with μ .”

- 3) Thirdly, we note that when QTE is not valid as pointed out by Melianas et al. (see Proc. Natl. Acad. Sci. U. S. A. 116, 23416–23425 (2019)), a different approach is needed. We added text in page 8:

“However in cases where QTE is no longer valid, for example if the spatially separated states are not strongly coupled, as explored by Melianas et al⁵², we would, in general, expect different chemical potential energies at different interfacial sites. Then the above expression for emission would need to be adapted as we explore in Supplementary Note 5.”

And we added the additional equations (Eq. (N5)) to model the case where different CT manifolds have different chemical potentials (μ_t) in Supplementary Note 5:

“This leads to a modified expression for the recombination rate constants and Z factor for recombination, as follows:

$$K_{nr} = \frac{1}{Z_{rec}} \sum_t \int_a^b \frac{2\pi}{\hbar} V_t(E_{CT})^2 FCWD_{rec,t}(0, E_{CT}) c_t D_t(E_{CT}) \exp\left(-\frac{E_{CT} - \mu_t}{k_B T}\right) d(E_{CT}), \quad (N5.1)$$

$$k_r(\hbar\omega) = \frac{1}{Z_{rec}} \sum_t \int_a^b \frac{1}{3\pi\epsilon_0\hbar^4} \left(\frac{\hbar\omega}{c}\right)^3 M_t(E_{CT})^2 FCWD_{rec,t}(\hbar\omega, E_{CT}) c_t D_t(E_{CT}) \exp\left(-\frac{E_{CT} - \mu_t}{k_B T}\right) d(E_{CT}), \quad (N5.2)$$

$$Z_{rec} = \sum_t \left\{ \sum_{m=0}^{\infty} \exp\left(-\frac{m\hbar\Omega_t}{k_B T}\right) \int_a^b c_t D_t(E_{CT}) \exp\left(-\frac{E_{CT} - \mu_t}{k_B T}\right) d(E_{CT}) \right\}. \quad (N5.3)$$

”

- 4) Lastly, regarding the state filling effect in injection dependent electroluminescence simulations, emission spectra change with increased injection (in both experiments in Fig. 5 and simulations in Fig. 6), which is a signature of state filling, and also indicates a departure from QTE under high injection. That means the emission spectra are no longer describable by the QTE theory as we introduced in the theory section, but in fact reflect a kinetically limited state filling effect, which could be described using Fermi-Dirac statistics (see PHYSICAL REVIEW B 86, 024201 (2012) and Adv. Energy Mater. 5, 1500123 (2015)). To make this clearer in the text, we have revised the relevant part in the main text and added additional discussion in supplementary note 4:

Page 16 in the main text:

“We first model the injection dependence of the EL spectrum (from Fig. 5) on $g(E_{CT})$ at different compositions (i.e. O-IDFBR wt%) by varying μ as described in in Supplementary Equation (N4.1). We note here that if the system obeyed QTE as assumed until now, the EL spectrum would not change shape under increasing injection (see Eq. (7b)). Since in fact the EL spectrum does change shape upon increasing injection, we need to use a different approach that takes into account the filling of states as previously proposed⁵² and as discussed in detail in Supplementary Note 4.^{9,65}”

Supplementary Note 4 in the SI:

“In the injection dependent electroluminescence simulations, the emission spectra are changing with increased injection (in both experiments in Fig. 5 and simulations in Fig. 6), which is a signature of a departure from quasi-thermal equilibrium (QTE) condition under high injection condition. That means the emission spectra are no longer describable by the QTE theory we introduced in the theory section (Eq. (7b)), but in fact reflect a kinetically limited state filling effect³¹. In previous work, Gong et al.³¹ addressed the problem by considering the bias dependent distributions of the electrons and holes that contribute to the EL, using Fermi-Dirac (FD) statistics. In a different approach, Burke et al.²² argued that the limited capacity of interface states for excitons meant that the CT states should obey FD statistics. Here, we adopt an approach similar to Burke and use the formalism below to model the EL data:

$$EL(\hbar\omega) = \frac{1}{Z_{rec}} \sum_t \int_a^b \frac{1}{3\pi\epsilon_0\hbar^4} \left(\frac{\hbar\omega}{c}\right)^3 M_t(E_{CT})^2 FCWD_{rec,t}(\hbar\omega, E_{CT}) c_t D_t(E_{CT}) \frac{1}{\exp\left(\frac{E_{CT} - \mu}{k_B T}\right) + 1} d(E_{CT}). \quad (N4.1)$$

”

Comment #3:

On page 5: What means they assume that radiative and non-radiative recombinations follow the energy gap law? How is this additional assumption included?

Responses:

We are sorry about the confusion caused, and have revised the text on page 5 to:

“As in previous models^{7,8,38}, we assume that radiative and non-radiative recombination occur only via the CT states, either directly after exciton dissociation or by reformation of the CT state from free charges. The radiative and non-radiative CT-to-ground-state transitions occur between vibronic modes of each state and are accompanied by the emission of a photon or of several vibrational quanta, respectively.^{43,44}”

Comment #4:

Why are the curves in Fig. 2a appear to be so wobbly? Is it because of the small linewidth? Please indicate the value of $\sigma_{CT,t}$ or λ that is causing this effect in the caption.

Responses:

Thank you for pointing this out. The effect is not actually caused by σ_{CT} nor λ , as Fig. 2a is just the visualization of the density distribution of CT states, which is the input and not the emission nor absorption spectrum. It appears to be just a numerical issue and has been fixed (see also below):

Fig. 2. General model results on the effect of static disorder.

Comment #5:

The authors talk about single Gaussian CT state distribution. Please clarify that you mean that the coefficients c_t are distributed like a Gaussian (if this is what is meant) and not that the resulting distribution of states is a Gaussian.

Responses:

Thank you for asking this question. For a single Gaussian CT state distribution, there is only one CT manifold, hence $g(E_{CT}) = c_1 D_1(E_{CT})$, and c_1 is a “constant” weighting factor not a distribution, instead $D_1(E_{CT})$ is the distribution function and of course the resulting distribution of states ($g(E_{CT})$) is Gaussian. We have also revised the text in line 99 to make it clearer:

“where c_t is a constant weighting coefficient such that $\sum_t c_t = 1$, and $D_t(E_{CT})$ is a line-shape function. Practically, $D_t(E_{CT})$ can be any function. In our model we use a gaussian line-shape for $D_t(E_{CT})$, i.e.

$$D_t(E_{CT}) = \frac{1}{\sigma_{CT,t}\sqrt{2\pi}} \exp \left[-\frac{1}{2} \left(\frac{E_{CT} - E_{CT,ct}}{\sigma_{CT,t}} \right)^2 \right], \text{ where } \sigma_{CT,t} \text{ is the width of}$$

the individual gaussian function.”

We sincerely hope that **Reviewer #2** is satisfied with our second-round responses, and that these changes made to the text have allowed us to assemble an improved and stronger manuscript.

REVIEWER COMMENTS

Reviewer #2 (Remarks to the Author):

The authors have carefully answered all my questions and the paper can now be accepted.